



# Effective radiative forcing of anthropogenic aerosols in E3SMv1: historical changes, causality, decomposition, and parameterization sensitivities

Kai Zhang[1], Wentao Zhang[2,1], Hui Wan[1], Philip J. Rasch[1], Steven J. Ghan[1], Richard C. Easter[1], Xiangjun Shi[2], Yong Wang[3], Hailong Wang[1], Po-Lun Ma[1], Shixuan Zhang[1], Jian Sun[1,10], Susannah Burrows[1], Manish Shrivastava[1], Balwinder Singh[1], Yun Qian[1], Xiaohong Liu[4], Jean-Christophe Golaz[5], Qi Tang[5], Xue Zheng[5], Shaocheng Xie[5], Wuyin Lin[6], Yan Feng[7], Minghuai Wang[8], Jin-Ho Yoon[9], and Ruby L. Leung[1]

[1]Pacific Northwest National Laboratory, Richland, WA, USA
[2]School of Atmospheric Sciences, Nanjing University of Information Science and Technology, Nanjing, Jiangsu, China
[3]Ministry of Education Key Laboratory for Earth System Modeling and Department of Earth System Science, Tsinghua University, Beijing, China
[4]Department of Atmospheric Sciences, Texas A&M University, College Station, Texas, USA
[5]Lawrence Livermore National Laboratory, Livermore, CA, USA
[6]Brookhaven National Laboratory, Upton, NY, USA
[7]Argonne National Laboratory, Lemont, IL, USA
[8]Nanjing University, Nanjing, China
[9]Gwangju Institute of Science and Technology, Gwangju, South Korea
[10]Now at National Center for Atmospheric Research, Boulder, Colorado, USA

**Correspondence:** K. Zhang (kai.zhang@pnnl.gov)

**Abstract.**

The effective radiative forcing of anthropogenic aerosols ($ERF_{aer}$) is an important measure of the anthropogenic aerosol effects simulated by a global climate model. Here we analyze $ERF_{aer}$ simulated by the E3SMv1 atmosphere model using both century-long free-running atmosphere-land simulations and short nudged simulations. We relate the simulated $ERF_{aer}$ to characteristics of the aerosol composition and optical properties, and evaluate the relationships between key aerosol and cloud properties.

In terms of historical changes from the year 1870 to 2014, our results show that the global mean anthropogenic aerosol burden and optical depth increase during the simulation period as expected, but the regional averages show large differences in the temporal evolution. The largest regional differences are found in the emission-induced evolution of the burden and optical depth of the sulfate aerosol: a strong decreasing trend is seen in the Northern Hemisphere high-latitude region after around 1970, while a continued increase is simulated in the tropics. Consequently, although the global mean anthropogenic aerosol burden and optical depth increase from 1870 to 2014, the $ERF_{aer}$ magnitude does not increase after around year 1970. The relationships between key aerosol and cloud properties (relative changes between preindustrial and present-day conditions) also show evident changes after 1970, diverging from the linear relationships exhibited for the period from 1870 to 2014.





The ERF$_{aer}$ in E3SMv1 is relatively large compared to the recently published multi-model estimates; the primary reason is the large indirect aerosol effect (i.e., through aerosol-cloud interactions). Compared to other models, E3SMv1 features a stronger sensitivity of the cloud droplet effective radius to changes in the cloud droplet number concentration. Large sensitivity is also seen in the liquid cloud optical depth, which is determined by changes in both the effective radius and liquid water path. Aerosol-induced changes in liquid and ice cloud properties in E3SMv1 are found to have a strong correlation, as the evolution

of anthropogenic sulfate aerosols affects both the liquid cloud formation and the homogeneous ice nucleation in cirrus clouds.

The ERF$_{aer}$ estimates in E3SMv1 for the shortwave and longwave components are sensitive to the parameterization changes in both liquid and ice cloud processes. When the parameterization of ice cloud processes is modified, the top-of-atmosphere forcing changes in the shortwave and longwave components largely offset each other, so the net effect is negligible. This suggests that, to reduce the magnitude of the net ERF$_{aer}$, it would be more effective to reduce the anthropogenic aerosol effect

through liquid or mixed-phase clouds.

## 1    Introduction

Variations in atmospheric composition play an important role in causing the observed (historical) climate changes since the pre-industrial era. The concentrations of both greenhouse gases and aerosols have changed, with partially offsetting impacts on the Earth's radiative balance. Compared to greenhouse gases, the lifetime of aerosols is much shorter and the processes involved in

the aerosol lifecycle are more complex. In addition, aerosols affect the Earth's radiative balance not only by directly scattering and absorbing shortwave and longwave radiation but also by acting as cloud condensation nuclei and ice nuclei and hence changing the cloud formation process and cloud radiative forcing. Consequently, aerosols have been considered to be a group of most uncertain forcing agents in the climate system (Penner et al., 2001; Forster et al., 2007; Boucher et al., 2013).

Due to our incomplete knowledge of the aerosol-related processes and the limited spatial and temporal resolutions used by

Earth system models or global climate models, most of such models have rather simplified representations of aerosol lifecycles and their interactions with clouds, precipitation, and radiation. The complex and multi-scale aerosol-related processes are simplified and represented differently in the models, leading to large difference in the simulated aerosol properties (Textor et al., 2006) and their impacts on clouds (Zhang et al., 2016). As a result, the estimated anthropogenic aerosol effects in global climate models are quite uncertain, both for the aerosol-radiation interactions (Schulz et al., 2006; Myhre et al., 2013) and

aerosol-cloud interactions (Quaas et al., 2009; Ghan et al., 2016).

In an earlier study, Golaz et al. (2019) reports that the present-day (average of 1995-2014) net (shortwave + longwave) effective radiative forcing of anthropogenic aerosols (ERF$_{aer}$) in the atmosphere component of the Energy Exascale Earth System Model version 1 (E3SMv1) is about -1.65 W m$^{-2}$. This value is outside the likely range of -0.63 to -1.37 W m$^{-2}$ estimated from models participating in the Radiative Forcing Model Intercomparison Project (RFMIP, Smith et al., 2020)

for the World Climate Research Programme Coupled Model Intercomparison Project-Phase 6 (CMIP6, Eyring et al., 2016). Furthermore, E3SMv1's present-day ERF$_{aer}$ is close to the lower bound of the multi-model estimate from Bellouin et al. (2020), who report a 68% likelihood range of -1.6 to -0.6 W m$^{-2}$ and a 90% likelihood range of -2.0 to -0.4 W m$^{-2}$. We note that





E3SMv1 considers complex interactions between aerosols, clouds, and radiation (see Section 2.1.2), so the estimated $ERF_{aer}$ is expected to be larger than those estimated in models with simplified aerosol treatment (Fiedler et al., 2019; Shi et al., 2019). The large aerosol forcing in E3SMv1 appears to be one of the reasons why the model cannot reproduce the observed global mean temperature evolution in the second half of the 20th century, as suggested by the analysis of Golaz et al. (2019) based on a two-layer energy balance model. The present study complements that analysis by providing a comprehensive investigation of the aerosol forcing characteristics.

This study evaluates the $ERF_{aer}$ simulated by the E3SMv1 Atmosphere Model (EAMv1, Rasch et al. (2019)) and relate the $ERF_{aer}$ to characteristics of the aerosol composition and optical properties. We also evaluate the relationships between key aerosol and cloud properties. Our analysis focuses on the following aspects:

- The historical changes of aerosol properties and the regional differences in $ERF_{aer}$ are analyzed in Section 3. We investigate whether the historical changes of aerosol properties are consistent with the changes in the aerosol forcing and explore the temporal variations in different regions.

- Second, following previous studies, we identify the most important aerosol, cloud, and radiation flux quantities and analyzed the relationship between them (Section 4). The analysis focuses on the relative changes of these quantities, based on the causality between them.

- Possible causes of the strong $ERF_{aer}$ in E3SMv1 is explored in Section 5. The decomposition method of Ghan et al. (2013) is used to quantify the contributions of individual forcing mechanisms (i.e., direct aerosol effect, indirect aerosol effect, and residual effect) to the top-of-atmosphere (TOA) and surface $ERF_{aer}$. The contribution of individual aerosol species to the overall $ERF_{aer}$ is quantified by perturbing one type of anthropogenic aerosol emissions at a time.

- The sensitivities of $ERF_{aer}$ to parameterization changes, including both formulation changes and parameter tuning, are discussed in Section 6. These parameterization changes were applied during the E3SMv1 development and some of them have been further tuned for an alternate configuration of E3SM (Ma et al., 2021).

- Section 7 evaluates the impact of anthropogenic aerosols on the near surface temperature through fast processes (in contrast to the slow processes involving ocean/sea ice responses to aerosol forcing).

The goal of the study is to better characterize and understand the strong $ERF_{aer}$ in E3SMv1, and hence provide insights to help constrain its value in future versions of the model. Descriptions of the model and simulations are provided in Section 2. The conclusions from the study are summarized in Section 8.

## 2 Model and simulations

The model used in this study is the atmospheric component of the E3SMv1, often referred to as EAMv1. A comprehensive description of EAMv1 can be found in Rasch et al. (2019) and the representation of aerosol processes is documented in Wang





et al. (2020), hence we only provides brief overviews here (Sections 2.1). The simulations conducted and analyzed in this study are described in Section 2.2.

## 2.1 Model description

### 2.1.1 Overview

EAMv1 uses the High-Order Methods Modeling Environment (HOMME) spectral element dynamical core (Taylor and Fournier, 2010; Dennis et al., 2012) and a cubed-sphere mesh. The standard model resolution is used, which has approximately 100 km horizontal resolution, with 30 spectral elements ("ne30"). The model has 72 vertical layers (see Figure 1 of Xie et al. (2018)) and a top at approximately 0.1 hPa, with about 21 levels below 700 hPa and 26 levels above 100 hPa . The key physical processes considered include deep convection (Zhang and McFarlane, 1995), turbulence, shallow convection, and cloud macrophysics that are handled by CLUBB (Golaz et al., 2002; Larson et al., 2002; Larson, 2017), cloud microphysics (Gettelman and Morrison, 2015; Wang et al., 2014), aerosol processes (see details in Section 2.1.2), and radiation (Iacono et al., 2008). More information about aerosol and cloud microphysics parameterizations is provided in the next subsection. Most parameterized physical processes use a time step of 30 min, which is also the time step for physics and dynamics coupling. Static (fixed sub-step length) or dynamic sub-stepping method is used by some parameterizations. For example, droplet activation and ice nucleation are coupled with CLUBB and cloud microphysical processes, with a coupling frequency of 5 min. Radiation time step is 1h. More details about process coupling and time stepping can be found in Zhang et al. (2018) and Wan et al. (2021b). The evaluations of simulated large-scale circulation and clouds in EAMv1 are documented in Rasch et al. (2019), Xie et al. (2018), and Zhang et al. (2019). EAMv1 is interactively coupled with a land surface model (Oleson et al., 2013), which also includes the dust emission calculation and the parameterization that considers the impacts of light-absorbing impurities (from BC and dust deposition) on surface snow/ice albedo (Wang et al., 2020).

### 2.1.2 Parameterization of aerosols and their interactions with radiation and clouds

Aerosol processes in EAMv1 (Wang et al., 2020) are represented by the Modal Aerosol Module (MAM) (Liu et al., 2012a, 2016). Major aerosol compositions are considered, including sulfate (SU), black carbon (BC), primary organic matter (POM), secondary organic aerosols (SOA), dust (DU), sea salt (SS), and marine organic aerosols (MOM). The aerosol size distribution is represented by four log-normal modes, with three soluble modes with mixed aerosol species and one primary carbon mode for BC and POM (see figure 2 of Wang et al. (2020)). Aerosols are assumed internally mixed within each mode, but externally mixed between different modes. MAM prognoses both the interstitial aerosols (particles suspended in the air and not in cloud droplets) and cloud-borne aerosols (particles in cloud droplets). The model evaluation of present-day aerosol properties in EAMv1 is documented in Wang et al. (2020). The model representation and evaluation of dust and marine organic aerosols are documented in Burrows et al. (2018) and Feng et al. (2021).

The aerosol optical properties are parameterized based on Ghan and Zaveri (2007), which approximates the calculation using analytic functions of the mode radius and refractive index of wet particles. The refractive indices for individual aerosol





compositions are from Hess et al. (1998), except that the refractive index for BC is taken from Bond and Bergstrom (2006).
The volume mixing rule is assumed when calculating the volume-mean wet refractive index for aerosol modes with mixed
compositions. The Köhler-theory-based water uptake scheme is used to calculate the wet particle size in each mode, based
on the predicted aerosol composition and hygroscopicity, as well as the relative humidity with respect to water (RHw) in the
clear-sky portion of the grid box. The hygroscopicity values for individual compositions can be found in Table 1 of Burrows
et al. (2018). In MAM, a RHw ceiling of 98% is applied for the aerosol water uptake calculation.

For liquid cloud formation, soluble aerosols can act as cloud condensation nuclei and get activated to form cloud droplets
in existing or newly-formed clouds. The Abdul-Razzak and Ghan (2000) activation scheme is used, which parameterizes the
aerosol activation (or droplet nucleation) based on the subgrid updraft velocity and the aerosol properties (especially for dry size
and hygroscopicity) in each mode. Instead of calculating the aerosol activation with a spectrum of updraft velocities as in Ghan
et al. (2001), MAM uses a so-called characteristic subgrid updraft velocity to simplify the calculation, which is parameterized
as a function of the turbulent kinetic energy (TKE) that is prognosed by CLUBB. A lower bound of $0.2\,\mathrm{m\,s^{-1}}$ is applied to
the characteristic subgrid updraft velocity, in order to compensate the potentially-underestimated turbulence strength due to
unresolved processes (e.g., cloud-top radiative cooling). The activation process is coupled with an explicit vertical diffusion
scheme with dynamical sub-stepping, which handles the vertical diffusion transport of all aerosol species and the cloud droplet
number. [1]

For mixed phase clouds, E3SMv1 uses a classical-nucleation-theory (CNT) -based parameterization. The deposition, im-
mersion, and contact freezing of natural dust and BC particles are considered based on a PDF (probability-density-function)
representation of contact angles for those particles (Wang et al., 2014). The initial ice particle size is assumed to be the same
as the mean cloud droplet size. Among the different heterogeneous ice formation mechanisms considered in E3SMv1 for
mixed-phase clouds, the immersion freezing of dust is the dominant source for ice formation. The contributions from other ice
nucleation pathways (the deposition and contact freezing of dust and the immersion freezing of BC) are much smaller. In ad-
dition, all the droplets are assumed to homogeneously freeze when T<-40°C (note that this is different from the homogeneous
nucleation of sulfate aerosols), with an initial size of $25\mu$m. The previously-used Meyers scheme (Meyers et al., 1992) for de-
position/condensation freezing and the contact nucleation scheme (Young, 1974) are optional and are explored as a sensitivity
test in this study (see Section 6).

For cirrus cloud formation (T<-37°C), the homogeneous freezing of sulfate aerosols and heterogeneous immersion freezing
on mineral dust are considered using the Liu and Penner (2005) ice nucleation scheme. For both nucleation mechanisms, an
initial ice particle size of $10\mu$m is assumed. As for droplet nucleation, the ice nucleation scheme also uses the characteristic
updraft velocity (as a function of TKE), with a lower bound of $0.2\,\mathrm{m\,s^{-1}}$. Sulfate aerosols (or sulfate solution droplets) in the
Aitken mode with diameter larger than a threshold are considered as ice-nucleating aerosols for the homogenous ice nucleation.
This parameter has large uncertainty and it is set differently in various models. For example, in the CAM5/CESM1 model, a
threshold size of $100\,\mathrm{nm}$ was used. While in some sensitivity studies (e.g., Liu et al., 2012b; Zhang et al., 2013; Shi et al.,

---

[1]CLUBB calculates the vertical diffusion transport of other quantities, including water vapor, cloud hydrometeors (excluding cloud droplet number), and
other trace gases.





2015), all Aitken mode particles are considered as potential ice-nucleating aerosols in cirrus clouds. For the E3SMv1 CMIP6 experiments and the reference model used in this study, a threshold size of 50 nm is used (Golaz et al., 2019). We discuss this

parameter sensitivity in Section 6.

## 2.2   Simulations

Four groups of simulations are analyzed in this study (see table 1). First, we further analyze the atmosphere-only simulations from Golaz et al. (2019) with prescribed historical SST and sea ice concentration for the years 1870-2014. In the reference simulation ensemble (AMIP), transient/historical anthropogenic and biomass burning/biofuel emissions as well as other forc-

ings (e.g., concentrations of green-house gases) are prescribed using the CMIP6 emission data (Hoesly et al., 2018; Feng et al., 2020). In the second simulation ensemble (AMIP1850aer), the anthropogenic and biomass burning/biofuel emissions are kept at 1850 level (also from CMIP6). Dimethyl sulfide (DMS) emissions for three time slices (1850, 2000, and 2100) are available from a coupled model simulation with detailed representation of DMS formation in sea water (Wang et al., 2018) and are linearly interpolated for other period. Dust, sea salt, and marine organic aerosol emissions are calculated interactively in all the

simulations analyzed in this study based on the simulated surface winds (frictional velocity for dust) and other surface quantities from the land and ocean components. We take the difference between AMIP and AMIP1850aer to obtain the temporal evolution of anthropogenic aerosol properties, including mass burden, cloud condensation nuclei (CCN) number, and optical depth, as well as the transient $ERF_{aer}$. Each set of simulations has 3 ensemble members and the ensemble-averaged fields are used for analysis.

The AMIP type simulations from Golaz et al. (2019) only include limited monthly mean output for aerosol-related quantities, so additional output data are needed for more detailed analysis. Due to limited computational resources, we rely on shorter simulations that are nudged towards reanalysis data for the additional three simulation groups. Nudged simulations help identify the aerosol forcing signal (i.e., spatial pattern) over shorter time periods (e.g, a specific year). See further discussions later in this section.

The second group includes nudged simulations with emissions of anthropogenic and biomass burning/biofuel aerosols and their precursors fixed to a specific year. We picked 6 time slices to represent preindustrial (E1850), early-industrial (E1900), mid-industrial (E1950), Europe-peak-industrial (E1970), near-present (E2000), and present (E2010, also as the nudged CTRL) conditions. By taking the difference between E1850 and all other simulations in this group, we derive the anthropogenic aerosol effects in each (emission) period and investigate the regional shifts of the aerosol forcing pattern. In addition to using

the CMIP6 emissions, we also performed simulations using the CMIP5 emissions (Lamarque et al., 2010). As shown in Table 2 and Figure E1, using the CMIP5 emissions results in a slightly smaller $ERF_{aer}$ estimate in E3SMv1.

    The third group consists of simulations with 1850 aerosol emissions, but with individual aerosol species the emission (or 3D production rates for SOA) set to present-day (2010) value. Again, by taking the difference between E1850 and simulations in this group, the aerosol forcing by individual aerosol species can be derived. For DMS emissions, the global mean difference

between emissions for 1850 and 2010 (interpolated) is pretty small (<1%), and sensitivity simulations show it has negligible impact on the simulated $ERF_{aer}$. Therefore, the result for DMS is not included here.





**Table 1.** List of simulation configurations. Note that for SOA, the 3D formation rates calculated based on Shrivastava et al. (2015) are applied as emissions in E3SMv1.

| Simulations | Description |
|---|---|
| **Group 1** | |
| AMIP | Free running simulation with prescribed SST, 1870-2014, with CMIP6 historical emissions of aerosols and their precursors |
| AMIP1850aer | As AMIP, but with CMIP6 pre-industrial (1850) emissions of aerosols and their precursors |
| **Group 2** | |
| CTRL (E2010) | Nudged simulation for year 2010, with year 2010 emissions of aerosols and their precursors |
| E1850 | as CTRL, but with year 1850 emissions of aerosols and their precursors |
| E1900 | as CTRL, but with year 1900 emissions of aerosols and their precursors |
| E1950 | as CTRL, but with year 1950 emissions of aerosols and their precursors |
| E1970 | as CTRL, but with year 1970 emissions of aerosols and their precursors |
| E2000 | as CTRL, but with year 2000 emissions of aerosols and their precursors |
| CMIP5E1850 | as CTRL, but with year 1850 emissions of aerosols and their precursors from CMIP5 |
| CMIP5E2000 | as CTRL, but with year 2000 emissions of aerosols and their precursors from CMIP5 |
| **Group 3** | |
| E2010BC | as E1850, but the anthropogenic BC emissions are for year 2010 |
| E2010POM | as E1850, but the anthropogenic POM emissions are for year 2010 |
| E2010SU | as E1850, but the anthropogenic SO2 and sulfate emissions are for year 2010 |
| E2010SOA | as E1850, but the anthropogenic SOA formation rates (applied as 3D emissions) are for year 2010 |
| E2010BB | as E1850, but the biomass burning (wild-fire) emissions for BC, POM, and SO2 are for year 2010. |
| **Group 4** | |
| BERG07 (PD and PI) | as CTRL or E1850, but the Bergeron process rate tuning parameter is changed from 0.1 to 0.7 following Ma et al. (2021) |
| KK2000 (PD and PI) | as CTRL or E1850, but the autoconversion tuning parameters are reset to Khairoutdinov and Kogan (2000) values |
| HOM100 (PD and PI) | as CTRL or E1850, but the cutoff size of Aitken mode sulfate aerosols for homogeneous ice nucleation is changed from 50 nm to 100 nm |
| MEY (PD and PI) | as CTRL or E1850, but the Meyers et al. (1992) heterogeneous ice nucleation scheme is used, instead of the CNT-based scheme (Wang et al., 2014). |

The last group includes simulations similar to the nudged CTRL and E1850, but with the cloud microphysics parameterization switched back to older schemes or tuned with a different parameter. Several pairs of simulations were performed with both present-day (PD) and pre-industrial (PI) aerosol emissions to estimate the anthropogenic aerosol effects.





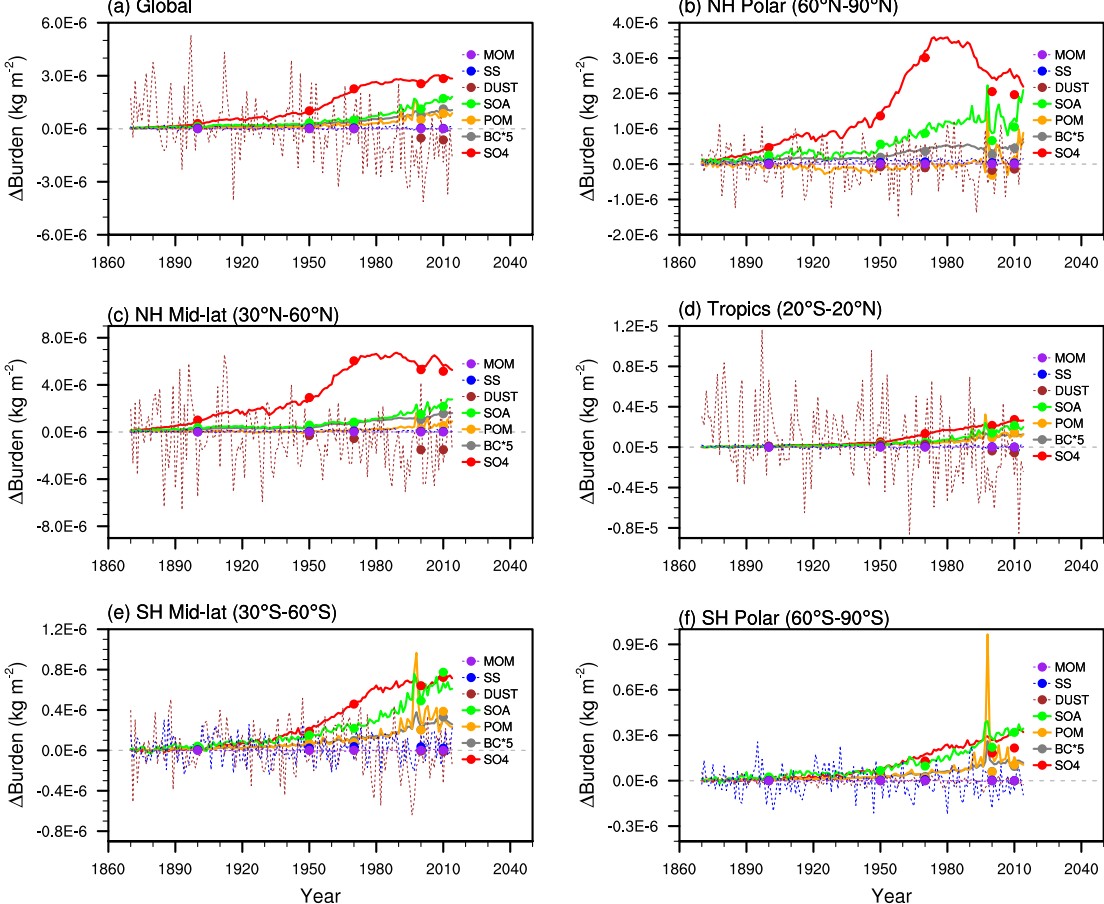

**Figure 1.** Temporal variations of global and annual mean aerosol burden differences between simulations with transient and pre-industrial (1850) aerosol emissions. Solid lines indicate aerosol species that are directly influenced by anthropogenic aerosol emissions (SO4, BC, POM, SOA) and dotted lines indicate natural aerosols (DUST, SS, MOM) that are mainly driven by surface winds. Dots indicate results from the short nudged simulations. Note that for clarity, the BC Δburden is scaled by a factor of 5.

Since the large-scale circulation is constrained, nudging can help distinguish the signals (e.g., caused by changes in external forcings) from the noises caused by chaotic model responses to forcing or parameterization changes (Kooperman et al., 2012; Zhang et al., 2014). We only nudge horizontal winds in our simulations, because it is better not to constrain the thermodynamical response of meteorological fields to aerosol effects. Also, earlier studies show negative impact of temperature and humidity nudging on the modeled climate (Zhang et al., 2014; Lin et al., 2016; Sun et al., 2019), which will make the forcing estimate inconsistent with that estimated using free-running simulations. For all the nudged runs, we nudge the horizontal winds towards the ERA-Interim reanalysis data, with a relaxation time scale of 6h. Since the vertical interpolation/extrapolation of reanalysis data to model vertical grids might have large errors near the surface, nudging is not applied in the lowest 2 layers. In the layer next to the no-nudging layers, the nudging relaxation time is doubled (weaker nudging compared to the layers above with full





nudging strength). Each nudged simulation is 15-month long and the results from a whole year (after a 3-month spin-up) are
used for analysis.

We note that since the horizontal winds are constrained in the nudged simulations, the anthropogenic aerosol effect on the large-scale circulation is greatly diminished. As we will show later, for global mean or regional mean (over a large area) estimates, the nudged simulations provide similar $\text{ERF}_{\text{aer}}$ estimates as the free-running simulations.

## 3   Historical changes and regional shifts of aerosol compositions, optical depth, and forcings

Before showing the effective aerosol forcing, we first analyze the simulated historical changes of *anthropogenic* aerosol burdens (vertical integration of aerosol mass) over the simulation period (1870-2014). Here, we define the *anthropogenic* change of a certain quantity (denoted as $\Delta$, e.g., aerosol burden, optical depth, and radiative fluxes) as the differences between the modeled quantities in simulations with transient historical forcing and the pre-industrial forcing (year 1850) for aerosols and their precursors. Note that for convenience, the anthropogenic change we define here also include the contribution from biomass
burning emissions.

### 3.1   Changes in anthropogenic aerosol burden and optical depth

Figure 1 shows the historical changes of global and regional mean anthropogenic aerosol burden ($\Delta$burden) for different compositions. In addition to the results from AMIP type simulations, we also show the nudged simulations for five selected years (shown as dots). Overall, the results from the nudged simulations agree very well with those from the AMIP type
simulations, both in terms of historical variations and regional differences of $\Delta$burden.

For aerosols with anthropogenic sources (i.e., sulfate, BC, POM, SOA), the global mean anthropogenic burden (Figure 1a) shows an overall increase during the simulation period (1870 to 2014). The burden of anthropogenic sulfate shows largest change, followed by that of SOA and POM (note that the BC $\Delta$burden is scaled by a factor of 10 in Figure 1). There are large regional differences in $\Delta$burden. For example, anthropogenic sulfate burden in the Northern Hemisphere (NH) polar
region (60°N-90°N) peaks at 1970s and decreases by about 50% in year 2010 compared to the peak value (Figure 1b). This is consistent with the emission reduction of air pollutants in Europe and North America where the legislation for clear air was implemented from the 1970s (Sliggers and Kakebeeke, 2004). While in the tropics (20°S-20°N), the sulfate burden continues to increase (Figure 1d).

The carbonaceous aerosols are not only affected by the anthropogenic aerosol sources (e.g., from industrial and transport
sectors), but also by the biomass burning emissions. In addition to the gradual increase in time, there are several intermittent spikes in the anthropogenic aerosol burden change. For example, the global biomass burning emission is largest in 1997 (van der Werf et al., 2017), an extreme El Nino year, and it contributes a large increase of carbonaceous aerosol burden change. The impact of biomass burning emissions is more evident in cleaner areas like SH mid-latitude and polar regions (Figure 1e,f). Similar increases are also present in the NH polar region, where biomass burning emissions increase significantly after 2011
(especially over Boreal North America and Boreal Asia, see figure 9 of van der Werf et al. (2017)).





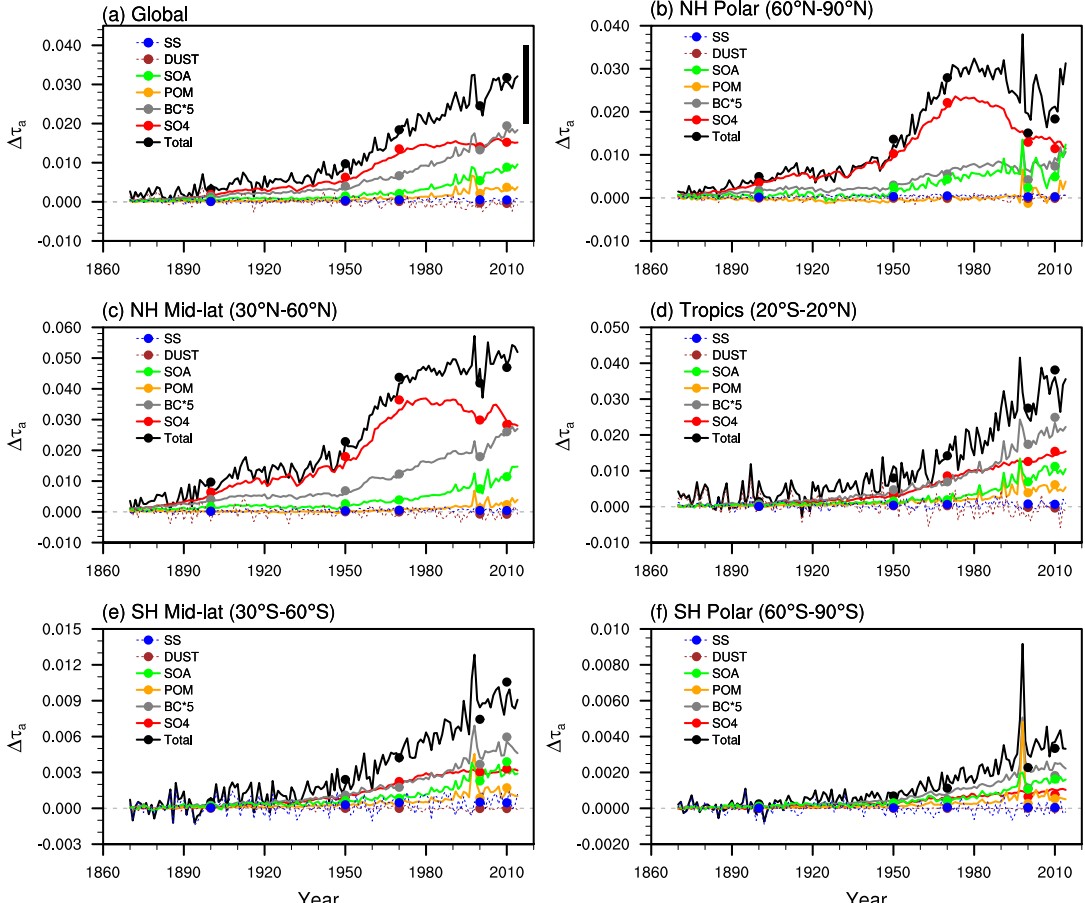

**Figure 2.** Similar to Fig 1, but for anthropogenic aerosol optical depth at $0.55\mu m$ (total and individual components). Solid lines indicate aerosol species that are directly influenced by anthropogenic aerosol emissions (SO4, BC, POM, SOA) and dotted lines indicate natural aerosols (DUST, SS, MOM) that are mainly driven by surface winds. Dots indicate results from the short nudged simulations. The black vertical bar in panel (a) shows the estimated range of total $\Delta\tau_a$ by Bellouin et al. (2020). Note that the BC values are scaled by a factor 5 for easier comparison. See section 3.1 for details.

The change in dust aerosol burden caused by the anthropogenic aerosol effect has large inter-annual variabilities and there is a small decreasing trend after about 1950, indicating that anthropogenic aerosols can indirectly affect the dust lifecycle (e.g., through the changes in surface winds or moisture), especially in the tropics and NH mid-latitude region (Figure 1d,c). Changes in sea salt and marine organic aerosol burdens are small (only evident in SH mid-latitude and polar regions) and also have large inter-annual variations.

Figure 2 shows the historical changes of global and regional mean anthropogenic aerosol optical depth ($\Delta\tau_a$) for the total and individual aerosol components. Again, we see consistent results between the estimates by AMIP type simulations and nudged simulations constrained to the same large-scale wind fields but with different years of emission input. The total $\Delta\tau_a$



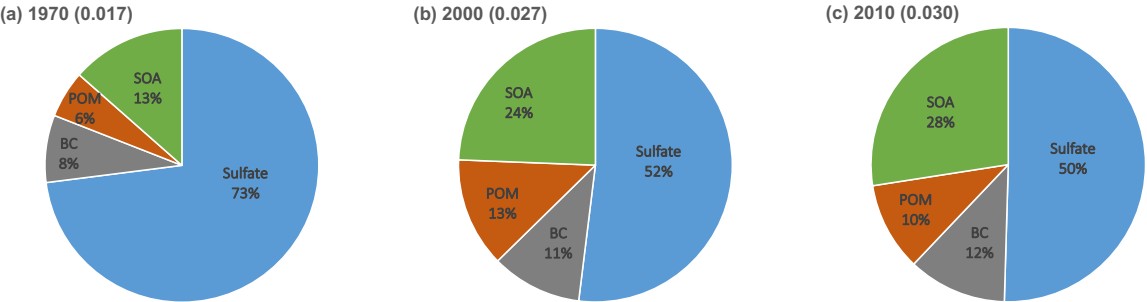

**Figure 3.** Contribution of sulfate, BC, POM, and SOA to anthropogenic aerosol optical depth (excluding dust and sea salt) in 1970, 2000, and 2010. Numbers in the panel title are the total anthropogenic aerosol optical depth ($\Delta\tau_{a,ant} = \Delta\tau_{a,SO4} + \Delta\tau_{a,BC} + \Delta\tau_{a,POM} + \Delta\tau_{a,SOA}$). Results are calculated using the AMIP simulations. For example, the contribution from sulfate is calculated as $\Delta\tau_{a,SO4}$ / $\Delta\tau_{a,ant}$. See section 3.1 for details.

(black) shows an overall increase during 1870-2014, but with large inter-annual variations. The inter-annual variability in total

$\Delta\tau_a$ is caused by changes in carbonaceous aerosols (BC, POM, SOA) affected by fluctuations in biomass burning emissions and by changes in dust aerosols. Compared to the multi-model estimates for $\Delta\tau_a$ (0.015-0.04) by Bellouin et al. (2020) based on multiple global model estimates (vertical bar in Figure 2a), the estimated present-day $\Delta\tau_a$ in E3SMv1 (about 0.03 around year 2010) is well within the range.

     $\Delta\tau_a$ of individual aerosol compositions shows similar temporal variations and regional differences as seen in $\Delta$burden

(Figure 1). It can be clearly seen that after 1970, both sulfate burden and sulfate $\Delta\tau_a$ over NH polar and NH mid-latitude regions decrease significantly, while for carbonaceous aerosols, there is an overall increase in burden and $\Delta\tau_a$. Figure 3 provides a more quantitative comparison of the contributions of sulfate, BC, POM, and SOA to the total anthropogenic $\Delta\tau_a$ (excluding dust and sea salt). The contribution of carbonaceous aerosols to total anthropogenic $\Delta\tau_a$ increases from 27% (BC 8%, POM 6%, and SOA 13%) in 1970 to 48% (BC 11%, POM 13%, and SOA 24%) in 2000, and 50% (BC 12%, POM 10%,

and SOA 28%) in 2010. Since carbonaceous aerosols are less hygroscopic compared to sulfate, their increased contribution to aerosol mass (burden) will decrease the mean hygroscopicity of the aerosol particles and thus affect the optical property and the ability to be activated to cloud droplets. As will be shown later, the changes of relative contribution of different types of anthropogenic aerosol components will have an impact on the relationship between $\Delta\tau_a$, CCN (cloud condensation nuclei) concentrations, and droplet concentrations (Nd).

**3.2   Historical changes in ERF$_{aer}$**

Figure 4 shows the historical changes of global and regional mean ERF$_{aer}$ ($\Delta$F) for the net (dark grey), shortwave (blue), and longwave (red) components. The global mean shortwave $\Delta$F is mostly negative and the largest negative forcing (about -2.6 W$m^{-2}$) appears first around 1980. In contrast, the longwave $\Delta$F is overall positive, with the largest forcing of about 1.0 W$m^{-2}$. After 1980, the global mean $\Delta$F doesn't show a clear increasing or decreasing trend, but there are fluctuations

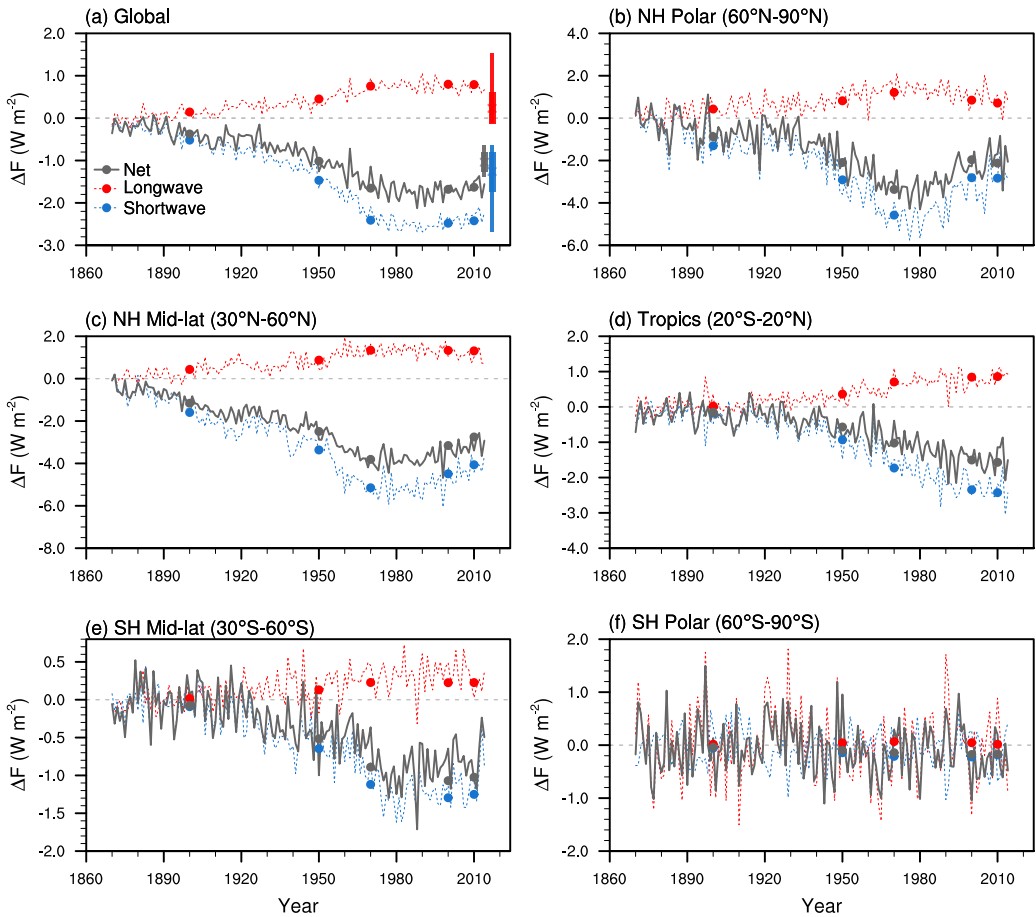

**Figure 4.** Temporal variations of global and annual mean $ERF_{aer}$ ($Wm^{-2}$) derived from the differences between simulations with transient and pre-industrial aerosol emissions. Solid lines indicate net $ERF_{aer}$ and dotted lines indicate the shortwave (blue) and longwave (red) components. Dots indicate results from the short nudged simulations. The cross, thick and thin vertical bars show the mean, standard deviations, and max/min $\Delta F$ estimated by CMIP6 RFMIP models (Smith et al., 2020). See section 3.2 for details.

caused by inter-annual variations. Similar to the $\Delta$burden and $\Delta\tau_a$ results, we can see large regional differences in the temporal evolution of $\Delta F$. The overall increasing or decreasing trend in $\Delta F$ resembles the changes in sulfate $\Delta$burden and $\Delta\tau_a$, especially for NH mid-latitude and polar regions.

Compared to $\Delta F$ estimated by CMIP6 RFMIP models (Smith et al., 2020), which adopt year 2014 emissions as present-day condition, the shortwave and longwave $\Delta F$ estimated by E3SMv1 are barely within the max-to-min range in the CMIP6 RFMIP model estimates (thin vertical bars), but outside of the one standard deviation around the mean estimate (thick vertical bars). While for the net $\Delta F$, E3SMv1 estimate is even larger than the max/min range of the multi-model estimates, since the large shortwave and longwave $\Delta F$ estimates from CMIP6 are from the same model and the net change is relatively small.



**Figure 5.** Geographical distributions of annual mean net (left), shortwave (middle), and longwave (right) $ERF_{aer}$ ($\Delta F$, W m$^{-2}$) estimated for different time slices using nudged simulations. Numbers shown on the top right of each panel are global mean values. For a given time slice, the CMIP6 aerosol emissions for the corresponding year are used. Simplified names ("E" removed) are used for the Group 2 listed simulations in table 1 and "2010" represents the CTRL simulation. See section 3.2 for details.

As for $\Delta$burden and $\Delta\tau_a$, the $\Delta F$ values estimated by the nudged simulations (dots in Figure 4) are very similar to the esti-mates from the AMIP-type free running simulations (lines). Therefore, in the following we mainly use results from the nudged simulations to discuss the regional forcing shifts, forcing decomposition (section 5), and the sensitivity to parameterization changes (section 6). Note that since the large-scale circulation is constrained, the estimated $ERF_{aer}$ is mainly affected by fast processes associated with interactions between aerosol, cloud, precipitation, and radiation.



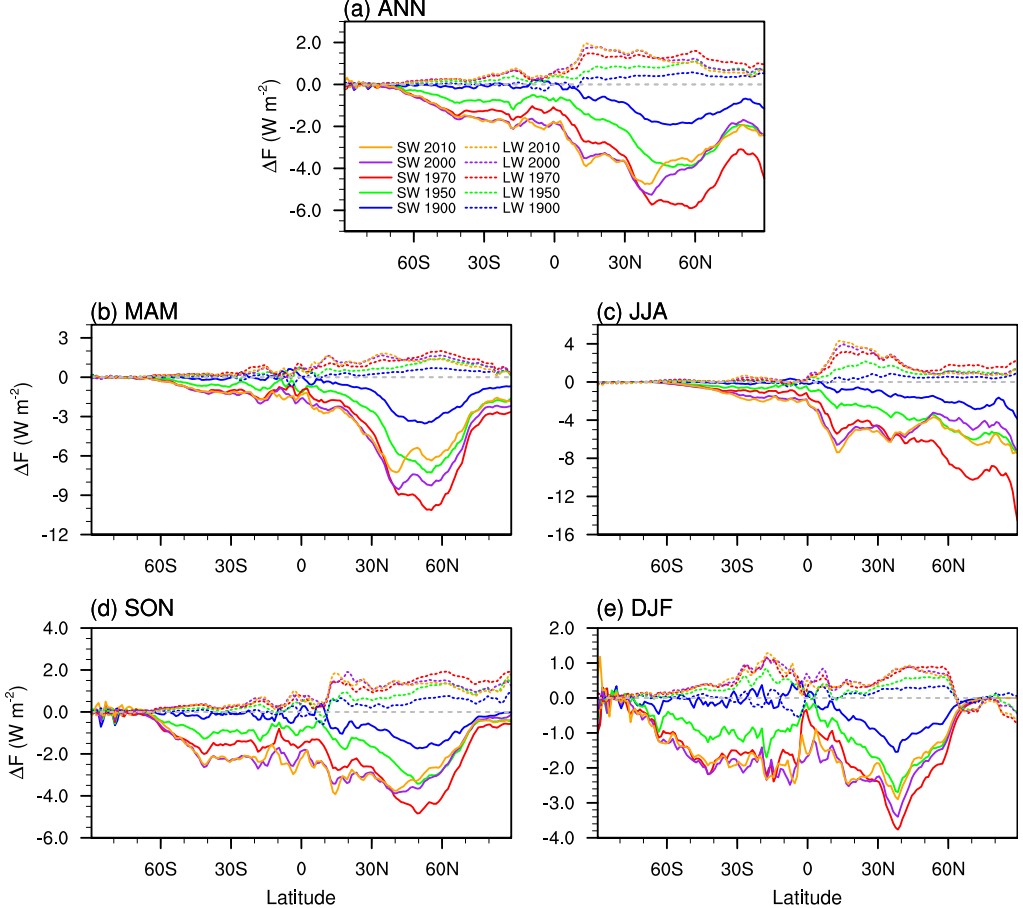

**Figure 6.** Annually and seasonally averaged zonal mean TOA $\Delta F$ (W m$^{-2}$) estimated for different time slices (emission years) indicated by different colors. Solid lines show the shortwave $\Delta F$ and dotted lines the longwave $\Delta F$. In JJA, the shortwave $\Delta F$ for 1970 (relative to 1850) is about -14 Wm$^{-2}$ near the North Pole. See section 3.2 for details.

Figure 5 shows the geographical distributions of $\Delta F$ for different time slices estimated based on several pairs of the nudged simulations (group 3 in table 1) with emissions set to 1850 or a specific year (1900, 1950, 1970, 2000, or 2010). The corre-

sponding changes in AOD and column-integrated number concentrations of CCN (at 0.1% supersaturation) and cloud droplets are shown in Figures H1, I1, and J1.

In 1900, relatively large $\Delta F$ appears over Western/Northern Europe and North America as well as the downwind regions. We see overall increase in the magnitude of $\Delta F$ in 1950 (about a factor of 3 larger than the 1900 value), but the regions with large $\Delta F$ are still very similar as in 1900. The $\Delta F$ over Western/Northern Europe and North America peaks in 1970. Afterwards, the

largest $\Delta F$ gradually shifts to East Asia. The estimated $\Delta F$ values for the years 2000 and 2010 are very similar, although the forcing over Western/Northern Europe and North America is further reduced in 2010.





Figure 6 clearly shows that $\Delta$F has strong seasonal variations. The global mean $\Delta$F is larger in boreal summer (Jun-Jul-Aug) and spring (Mar-Apr-May), mainly due to stronger oxidations of $SO_2$ and higher sulfate concentrations in these seasons. Also, both the annual and seasonal averaged zonal mean $\Delta$F fields have a regional shift of larger aerosol forcing from NH
high-latitude regions in 1970 to the tropics and subtropics in 2000 and 2010, mainly due to regional changes in anthropogenic emissions. The largest changes in NH polar region appear in boreal summer, from about -8 $\mathrm{Wm}^{-2}$ in 1970 to -4 $\mathrm{Wm}^{-2}$ in 2010.

In winter, shortwave $\Delta$F in the NH polar regions is close to zero due to very small incoming shortwave radiation. As is shown later, the increased liquid water path by anthropogenic aerosol effects results in a positive (longwave) radiative forcing
at the surface, which causes surface temperature to increase in addition to the greenhouse warming.

## 4 Relationships between changes in aerosol, cloud, and radiative flux quantities

The impacts of aerosol perturbations on the TOA radiative flux changes involve many processes related to aerosol and cloud microphysics. Previous studies (Quaas et al., 2009; Ghan et al., 2016; Bellouin et al., 2020) have summarized the key causal relationships that determine the aerosol effect through interactions with radiation and clouds. These relationships are often
expressed in the logarithmic form of dlnY/dlnX, since most processes involved in aerosol-cloud interactions are more sensitive to a relative change (between pre-industrial and present-day conditions) of a quantity, instead of the absolute change (Carslaw et al., 2013; Bellouin et al., 2020). We follow the conventional expressions as used in Ghan et al. (2016) and Bellouin et al. (2020). In the following discussions, $\Delta \ln X$ is the relative change of X (spatial and temporal mean) between PD and PI simulations and it is calculated as [2] $\Delta X/X_{\mathrm{PD}}$.

In earlier studies, some aerosol and cloud quantities are sampled at certain locations or under selected conditions, based on their role in the physical processes or the consideration for comparing with satellite retrievals. For example, the cloud-base values of CCN (e.g., 1 km above the surface) and the cloud top values of cloud droplet number concentrations ($N_d$) are often used (e.g., Quaas et al., 2009; Ghan et al., 2016; Zhang et al., 2016). Furthermore, to focus on the analysis on certain types of clouds, in some studies (e.g., Ghan et al., 2016; Gryspeerdt et al., 2020) these quantities are conditionally sampled only when
some criteria are met (e.g., when cloud top temperature is larger than a threshold value). For such analysis, high-frequency output data for the related aerosol and cloud properties are needed, unless the model can conditionally sample data online and process them accordingly (Wan et al., 2021a).

It should be noted that in our AMIP simulations, not all the quantities needed for such analysis are available or available at the required frequency (e.g., 3-hourly). Therefore, for some quantities (such as for CCN and $N_d$), we use the available column-
integrated quantities sampled as monthly mean fields for the analysis using AMIP simulations. For the nudged simulations, both monthly-mean output and high-frequency data sampled under required conditions (as in Ghan et al. (2016)) are available, so we use both of the two and evaluate whether the results are sensitive to the data sampling methods.

---

[2]If $X_{\mathrm{PI}}$ is used instead of $X_{\mathrm{PD}}$, $\Delta \ln X$ will be smaller for most quantities, but the relative differences between the simulations won't change.



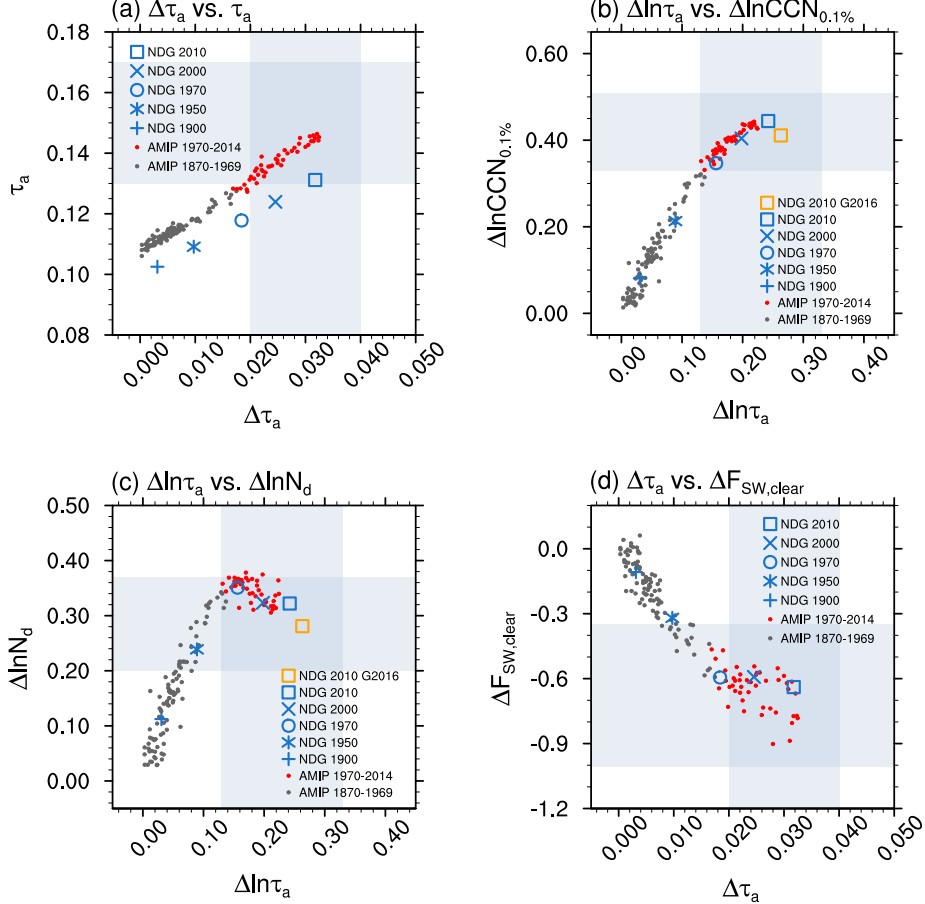

**Figure 7.** Relationships between simulated anthropogenic AOD ($\Delta\tau_a$) and present-day total AOD ($\tau_a$), relative (PD-PI) changes in AOD ($\Delta\ln\tau_a$) and in the column-integrated CCN at 0.1% supersaturation ($\Delta\ln CCN_{0.1\%}$), $\Delta\ln\tau_a$ and relative changes in the column-integrated $N_d$ ($\Delta\ln N_d$), $\Delta\tau_a$ and changes in clear-sky shortwave radiation ($\Delta F_{SW,clear}$). $\Delta\ln X = \Delta X/X_{PD}$. Dark grey dots indicate estimated values from AMIP PD/PI simulations for years 1870-1969, while red dots are the estimated values for years 1970-2014. Each dot represents the ensemble-averaged global annual mean field for a certain year. Blue markers show the estimates based on nudged simulations. The orange square shows the nudged simulation results averaged from conditionally sampled high-frequency data following Ghan et al. (2016), where CCN (at 0.3% supersaturation) and $N_d$ are sampled below 1 km and at the top of liquid clouds. Grey areas in panels (a,d) indicate the multi-model estimated range from Bellouin et al. (2020), and in panels (b,c) the multi-model estimated range from Ghan et al. (2016). See section 4.1 for details.

## 4.1 Relationships between $\Delta\ln\tau_a$ (or $\Delta\tau_a$) and other quantities

Figure 7 shows the relationships between simulated changes in AOD ($\Delta\tau_a$, $\Delta\ln\tau_a$) and key cloud and radiative flux properties.
The global annual mean total AOD ($\tau_a$, Figure 7a) from the AMIP simulation increased (almost linearly) from about 0.110 to 0.145 during the simulation period. Compared to the AMIP simulation (small dots), $\tau_a$ is consistently smaller in the nudged



simulations with anthropogenic emissions for different years (shown as bigger markers). This is mainly caused by the slightly weakened dust emission when reanalysis nudging is applied (Timmreck and Schulz, 2004; Sun et al., 2019). Although we don't apply nudging in the surface layer, the simulated surface winds and dust emission fluxes are still affected by the nudging

in upper layers (see Figure 18 in Sun et al. (2019)). The estimated present-day (year 2010) $\tau_a$ is well within the range of multi-model estimates (0.12-0.16 from Ghan et al. (2016) and 0.13-0.17 from Bellouin et al. (2020), shaded area in Figure 7). The simulated anthropogenic AOD ($\Delta\tau_a$) shows a similar increase during the simulation period. The estimated present-day $\Delta\tau_a$ ($\sim$0.03) in E3SM is also well within the range of multi-model estimates.

The relationship between $\Delta\ln\tau_a$ ($\Delta\tau_a/\tau_a$) and $\Delta\ln CCN_{0.1\%}$ is shown in Figure 7b. They have a quasi-linear correlation

before 1970 (grey dots). As indicated by the changed slope, the increase in $\Delta\ln CCN_{0.1\%}$ is weakened as $\Delta\ln\tau_a$ further increases after 1970 (red dots). Results from the nudged simulations show very similar changes. When sampling the data conditionally (cloud top temperature > -10°C) at high frequency, the result (shown as orange square in Figure 7b,c,d) is similar to that derived using the monthly mean data available and it is within the multi-model estimated range in Ghan et al. (2016). $\Delta\ln\tau_a$ and $\Delta\ln N_d$ also show a quasi-linear correlation before 1970, but the slope changes sign for samples after 1970. This

is likely relevant to the fact that there is a continuous increase of anthropogenic carbonaceous aerosols in most regions (that increases $\tau_a$) but decreased or stabilized anthropogenic sulfate aerosols (that decreases $N_d$) in the NH high- and mid-latitude regions. We note that the $N_d$ changes in response to CCN changes are different between individual regions (see Figures I1 and Figure J1 in Appendix, and discussions in Section 4.2). In addition, the droplet nucleation is also dependent on the maximum supersaturation in a rising cloud parcel, which is parameterized based on the ambient meteorological conditions (e.g., vertical

mixing and subgrid updraft velocity).

Following Bellouin et al. (2020), Figure 7d shows the relationship between $\Delta\tau_a$ and the changes in clear-sky shortwave radiation ($\Delta F_{SW,clear}$). $\Delta F_{SW,clear}$ magnitude increases with increasing $\Delta\tau_a$ almost linearly, but the increase is much weaker after 1970. This is likely also related to the fact that there are relatively more carbonaceous aerosols but less/stabilized sulfate aerosols, which increase the absorption and offset the cooling effect by scattering. The estimated $\Delta F_{SW,clear}$ in E3SMv1 is

well within the multi-model estimates presented in Bellouin et al. (2020).

## 4.2 Susceptibility of $N_d$, $LWP$, and $R_{e,liq}$ to changes in CCN

Figure 8 shows the relationships between simulated relative changes in column-integrated CCN at 0.1% supersaturation ($\Delta\ln CCN_{0.1\%}$) and in cloud microphysical quantities, including cloud droplet number concentration ($\Delta\ln N_d$), liquid water path of stratiform clouds ($\Delta\ln LWP_{strat}$), and cloud-top droplet effective radius ($\Delta\ln Re_{liq}$). The global annual mean $CCN_{0.1\%}$

increases by more than 80% from pre-industrial era to present-day condition. Compared to year 1970, the global annual mean $\Delta CCN_{0.1\%}$ increases by more than 60% in year 2010. However, it shows opposite trends in the NH polar region and in the tropics (see Figure A1 and C1 in Appendix).

Before 1970, there is strong positive correlation between $\Delta\ln N_d$ and $\Delta\ln CCN_{0.1\%}$ (grey dots in Figure 8b) and between $\Delta\ln LWP_{strat}$ and $\Delta\ln CCN_{0.1\%}$ (Figure 8c), but negative correlation between $\Delta\ln Re_{liq}$ and $\Delta\ln CCN_{0.1\%}$ (Figure 8d). This is

consistent with the causal relationship between these quantities simulated in the model (more CCN leads to higher $N_d$ and LWP,



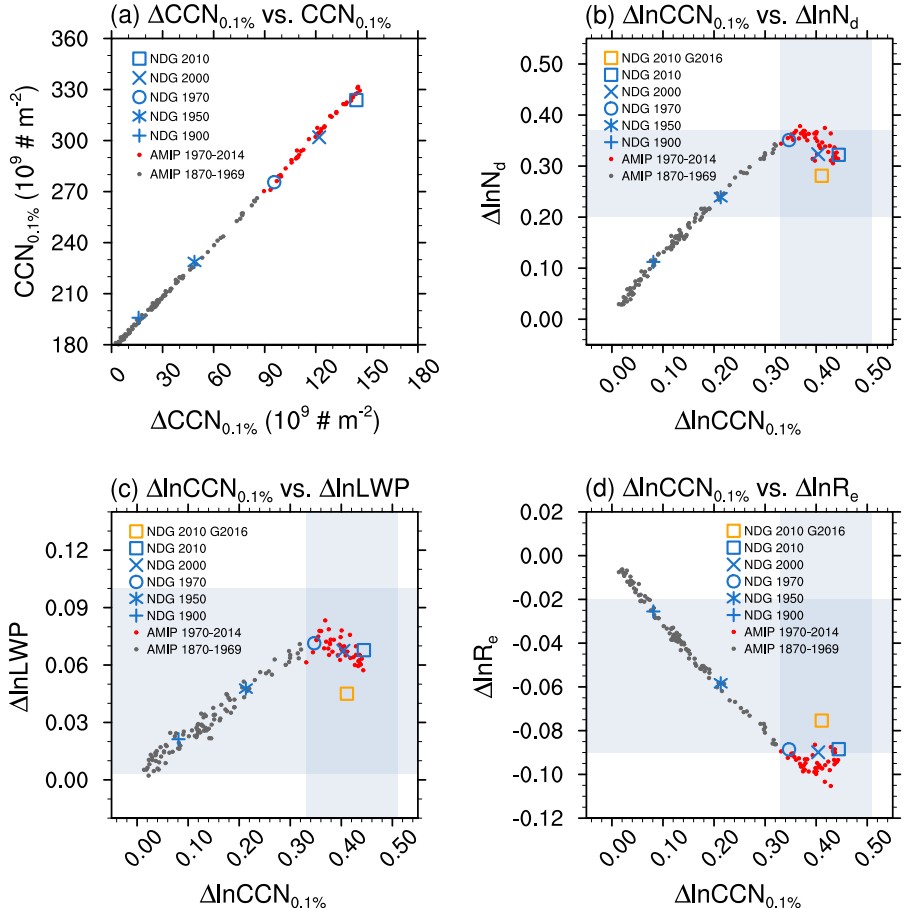

**Figure 8.** As Figure 7, but for relationships between the relative changes in CCN ( $\Delta lnCCN_{0.1\%}$) and in liquid cloud properties including droplet number ($\Delta lnN_d$), liquid water path ($\Delta lnLWP$), and effective radius of cloud droplets at cloud top ($\Delta lnRe_{liq}$). $\Delta lnX = \Delta X/X_{PD}$. Dark grey dots indicate estimated values from AMIP PD/PI simulations for years 1870-1969, while red dots are the estimated values for years 1970-2014. Each dot represents the ensemble-averaged global annual mean field for a certain year. Blue markers show the estimates based on nudged simulations. The orange square shows the nudged simulation results averaged from conditionally sampled high-frequency data following Ghan et al. (2016), where CCN (at 0.3% supersaturation) and $N_d$ are sampled below 1 km and at the top of liquid clouds. Grey area indicates the multi-model estimated range from Ghan et al. (2016). See section 4.2 for details.

but smaller $Re_{liq}$) and the fact that all types of anthropogenic aerosol emissions increase (so does CCN) from preindustrial era to about 1970 (see Figure 1). After 1970, the correlation between these global quantities become weaker (red dots in Figure 8b-d). This is mainly caused by two factors. First, more primary (carbonaceous) aerosols and less secondary (sulfate) aerosols decrease the particle hygroscopicity and size, which subsequently reduce the droplet nucleation even if $CCN_{0.1\%}$ doesn't change much
(e.g., Figure A1b and Figure B1b in Appendix). Second, there are regional differences in the simulated relationships. For





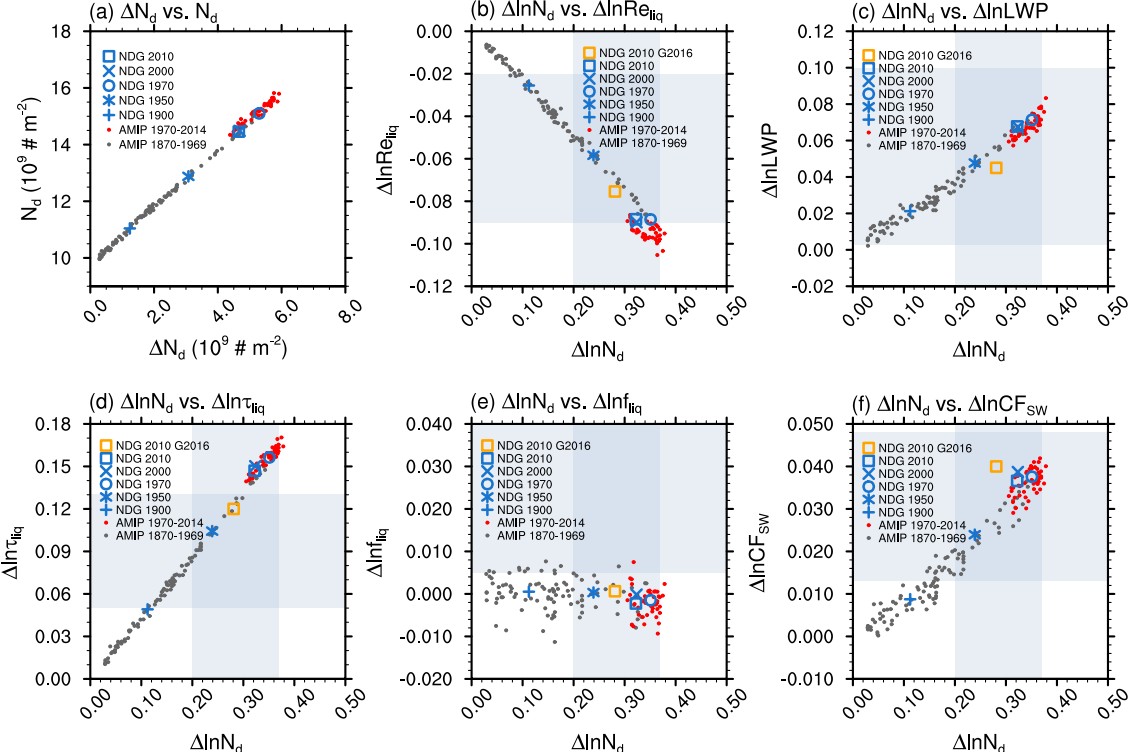

**Figure 9.** As Figure 7, but for relationships between $\Delta N_d$ and $N_d$, and between relative changes in $N_d$ and other quantities. $\Delta \ln \tau_{\mathrm{liq}}$, $\Delta \ln f_{\mathrm{liq}}$, and $\Delta \ln \mathrm{CF}_{\mathrm{SW}}$ are the relative changes in liquid cloud optical depth, liquid cloud fraction, and shortwave cloud radiative forcing respectively. $\Delta \ln \mathrm{X} = \Delta \mathrm{X}/\mathrm{X}_{\mathrm{PD}}$. Dark grey dots indicate estimated values from AMIP PD/PI simulations for years 1870-1969, while red dots are the estimated values for years 1970-2014. Each dot represents the ensemble-averaged global annual mean field for a certain year. Blue markers show the estimates based on nudged simulations. The orange square shows the nudged simulation results averaged from conditionally sampled high-frequency data following Ghan et al. (2016), where CCN (at 0.3% supersaturation) and $N_d$ are sampled below 1 km and at the top of liquid clouds. Grey area indicates the multi-model estimated range from Ghan et al. (2016). See section 4.3 for details.

example, in the NH polar region, $\mathrm{CCN}_{0.1\%}$ increases before 1970 but decreases afterwards (Figure A1a in Appendix). While in the tropics, $\mathrm{CCN}_{0.1\%}$ increases continuously during the whole simulation period (Figure C1a in Appendix).

Compared to the multi-model estimates reported in Ghan et al. (2016), the estimated values of $\Delta \ln \mathrm{CCN}_{0.1\%}$, $\Delta \ln N_d$, and $\Delta \ln \mathrm{LWP}_{\mathrm{strat}}$ in E3SMv1 are all well within the min-to-max range (almost right in the middle). However, we find the estimated

$\Delta \ln \mathrm{Re}_{\mathrm{liq}}$ magnitude in E3SMv1 is relatively large compared to the multi-model estimates.

### 4.3  Susceptibility of $\mathrm{Re}_{\mathrm{liq}}$, LWP, $\tau_{\mathrm{liq}}$, and cloud radiative forcing to changes in $\mathrm{N}_d$

Figure 9 shows the relationships between simulated relative changes in column-integrated $N_d$ ($\Delta \ln N_d$) and in other cloud microphysical quantities, including cloud-top droplet effective radius ($\Delta \ln \mathrm{Re}_{\mathrm{liq}}$), liquid and ice water path of stratiform clouds





($\Delta\ln$LWP and $\Delta\ln$IWP), cloud optical depth ($\Delta\ln\tau_{\mathrm{liq}}$), liquid cloud fraction ($\Delta\ln f_{\mathrm{liq}}$), and shortwave cloud radiative forcing

($\Delta\ln$CF$_{\mathrm{SW}}$).

The global annual mean $N_d$ increases by about 50% from pre-industrial era to present-day condition (blue square in Figure 9a). Compared to year 1970, the global annual mean $N_d$ is smaller in year 2010, which is different from the results we see in $\tau_a$ (Figure 7a) and CCN$_{0.1\%}$ (Figure 8a). More aerosols simulated in the model (in terms of global annual mean) do not necessarily lead to increase in $N_d$, since there are regional decreases in the sulfate burden that reduce the aerosol hygroscopic-

ity and subsequent aerosol activation. On the other hand, similar opposite trends (as for $\Delta\ln$CCN$_{0.1\%}$) appear in the NH polar region (decreases after 1970) and in the tropics (increase continuously after 1970).

There is a strong negative correlation between $\Delta\ln N_d$ and $\Delta\ln$Re$_{\mathrm{liq}}$ (grey and red dots in Figure 9b). There is also strong positive correlation between $\Delta\ln N_d$ and other quantities including $\Delta\ln$LWP, $\Delta\ln\tau_{\mathrm{liq}}$, and $\Delta\ln$CF$_{\mathrm{SW}}$ (Figure 9c,d,f). The results are consistent with the causal relationship between these quantities simulated in the model (more $N_d$ leads to larger

LWP, but smaller Re$_{\mathrm{liq}}$). Unlike $\Delta\ln$CCN$_{0.1\%}$, after 1970, the correlation between $N_d$ and other global quantities is very similar to those before 1970.

### 4.4 Relationships between changes in liquid and ice cloud properties

As described in section 2.1.2, E3SMv1 considers the activation of aerosol particles to cloud droplets and the freezing of cloud droplets to ice crystals in mixed-phase clouds when ice nuclei are available. It also considers the homogeneous ice nucleation

of sulfate aerosols and heterogeneous ice nucleation of dust aerosols in cirrus clouds. Because dust is natural and the ice nucleation rate caused by BC is rather small (compared to the contribution by dust), sulfate is the dominant anthropogenic aerosol type that affects both the droplet (and the subsequent freezing) and ice crystal formation processes.

Figure 10 shows the relationships between simulated relative changes in liquid and ice cloud properties. There are strong positive correlations between the changes in the liquid and ice cloud quantities for the global annual mean column-integrated

particle number ($\Delta\ln N_d$ vs. $\Delta\ln N_i$), cloud-top effective radius ($\Delta\ln$Re$_{\mathrm{liq}}$ vs. $\Delta\ln$Re$_{\mathrm{ice}}$), column-integrated water path in stratiform clouds ($\Delta\ln$LWP vs. $\Delta\ln$IWP), and the cloud optical depth ($\Delta\ln\tau_{\mathrm{liq}}$ vs. $\Delta\ln\tau_{\mathrm{ice}}$). The *ice* properties shown here don't consider the contribution from snow. On the other hand, the relative changes in liquid and ice cloud fractions are not well-correlated. Compared to liquid clouds, the ice cloud properties have a similar but slightly weaker sensitivity to changes in anthropogenic aerosols (Figure D1).

The relative change in global annual mean column-integrated droplet number concentration ($\Delta\ln N_d$ in Figure 10a, also in Figure 9) increases from the pre-industrial era, but starts to decrease after 1970. This is not the case for the relative changes in the column-integrated ice crystal number, where we see a continuous increase. This difference is mainly caused by the fact that the major changes in $N_d$ and $N_i$ appear in different regions. As shown in Figure 11, the largest changes in $N_d$ are in NH mid-latitude and polar regions, where anthropogenic aerosols affect the low liquid cloud most. In contrast, the largest changes

in $N_i$ are mainly in the tropics and sub-tropics regions, where the cirrus clouds are affected by anthropogenic sulfate particles. The anthropogenic aerosol effects on $N_i$ below 300hPa (mainly in mixed-phase clouds) are very small.





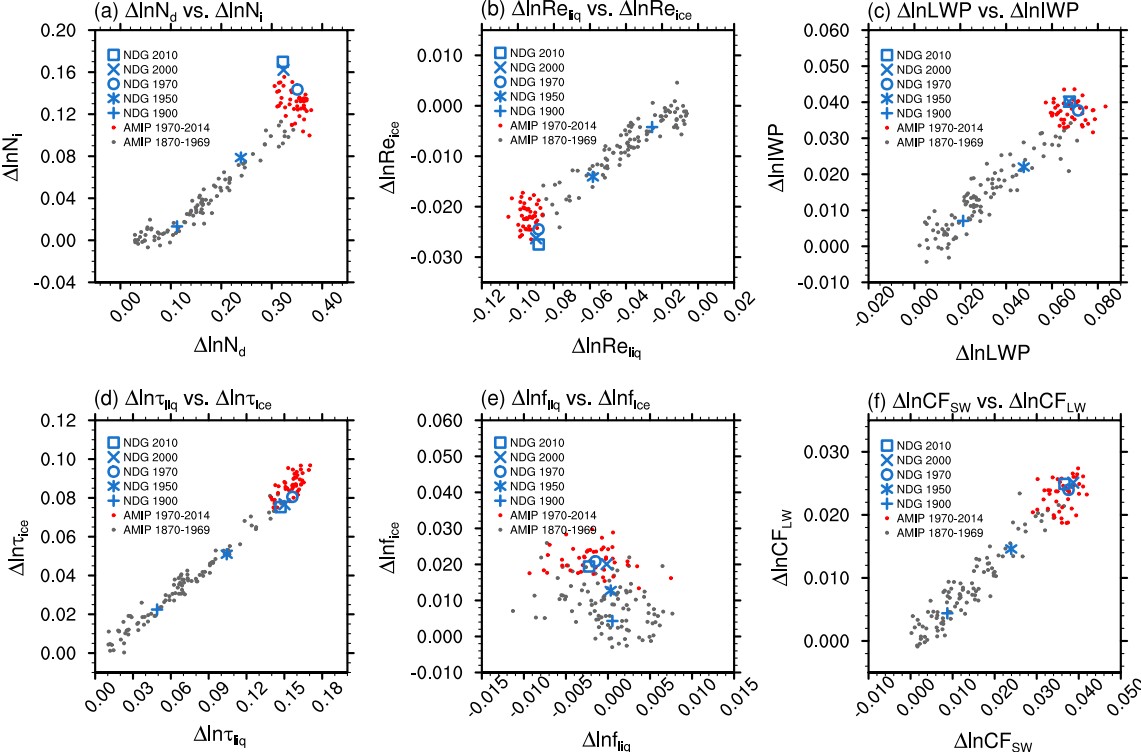

**Figure 10.** As Figure 7, but for relationships between changes in liquid and ice cloud properties as well as in SW and LW cloud radiative forcing. $\Delta\ln Re_{ice}$, $\Delta\ln IWP_{ice}$, $\Delta\ln\tau_{ice}$, $\Delta\ln f_{ice}$, and $\Delta\ln CF_{LW}$ are the relative changes in the effective radius of cloud ice at the cloud top, ice water path in stratiform clouds, ice cloud optical depth, ice cloud fraction, and longwave cloud radiative forcing respectively. $\Delta\ln X = \Delta X / X_{PD}$. Dark grey dots indicate estimated values from AMIP PD/PI simulations for years 1870-1969, while red dots are the estimated values for years 1970-2014. Each dot represents the ensemble-averaged global annual mean field for a certain year. Blue markers show the estimates based on nudged simulations. See section 4.4 for details.

## 5 Forcing decomposition

In this section, we decompose the effective aerosol forcing based on the method proposed by Ghan et al. (2013) and based on the individual aerosol forcing/emission types. These decompositions help us understand the role of various forcing mechanisms and the impacts of individual aerosol compositions on the overall $ERF_{aer}$.

### 5.1 Decomposition of direct, indirect, and surface albedo effect

We use the Ghan et al. (2013) method to decompose the total $ERF_{aer}$ ($\Delta F$) into individual forcings caused by aerosol-radiation interactions (direct aerosol effect), aerosol-cloud interactions (indirect aerosol effect), and the residual term (surface albedo effect). To do this, an additional call to the radiation calculation is added in the model, to calculate the radiative fluxes without

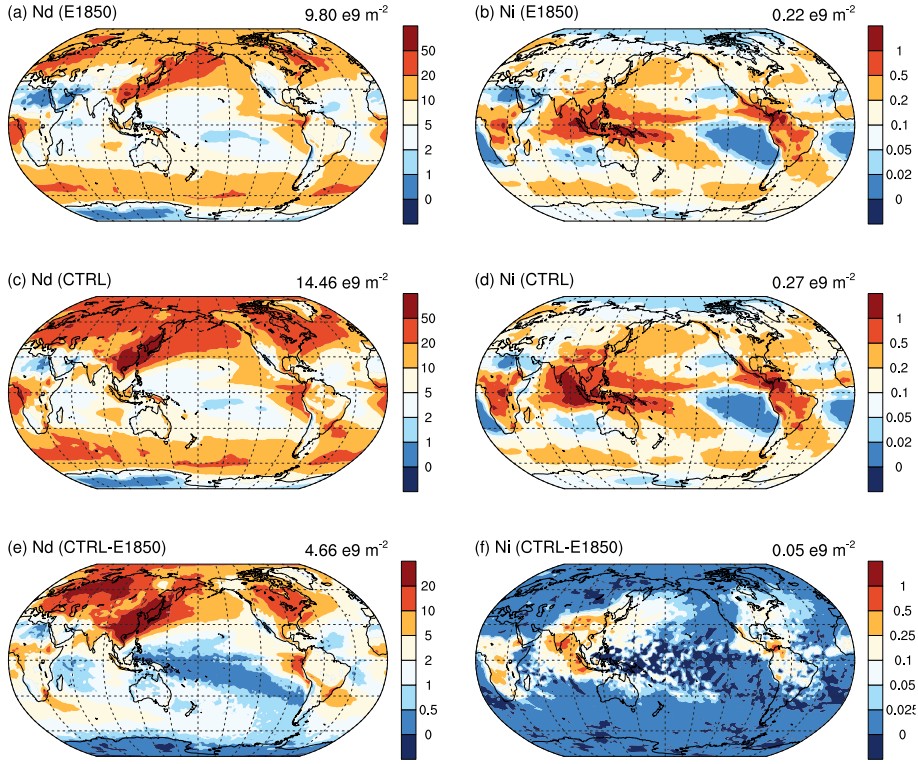

**Figure 11.** Annual mean column-integrated cloud droplet and ice crystal number concentrations for pre-industrial (E1850, panel a,b), present-day (CTRL for year 2010, panel c,d), and the PD-PI differences (e,f) in the nudged simulations. See section 4.4 for details.

considering the aerosol scattering and absorption of shortwave radiation in the calculation for all-sky ($F_{clean}$) and clear-sky ($F_{clear,clean}$) conditions. The decomposed forcings are calculated as follows:

– Aerosol-radiation interaction ("direct radiative forcing" in Ghan et al. (2013) ): $\Delta(F - F_{clean})$

– Aerosol-cloud interaction ("cloud radiative forcing" in Ghan et al. (2013) ): $\Delta(F_{clean} - F_{clear,clean})$

– Residual forcing ("surface albedo forcing" in Ghan et al. (2013) ): $\Delta F_{clear,clean}$

Figure 12 shows the geographical distributions of the decomposed TOA $\Delta F$ for the year 2010 using the nudged simulations with E3SMv1 (CTRL - E1850). The forcing caused by aerosol-cloud interactions (indirect effect) has the largest contribution to the overall $\Delta F$. In contrast, the global annual mean forcing caused by aerosol-radiative interactions is very small ($0.04\,\mathrm{Wm^{-2}}$), but it is positive over sub-tropical land and stratocumulus-prevalent regions, where BC absorption of shortwave radiation is strong (especially above clouds). The clear-sky aerosol forcing is overall negative, except over the North Africa and the Arabian
Peninsula, where there are significant changes in absorbing aerosols. The residual forcing ($\Delta F_{clear,clean}$) is negative over the NH polar region, which is mainly caused by anthropogenic sulfate aerosols. Positive forcing appears over subtropical land areas and it is caused by both shortwave (surface albedo effect) and longwave (water vapor) changes.





**Figure 12.** Geographical distributions of decomposed net (left column), shortwave (middle column), and longwave (right column) $\Delta F$ at TOA. The decomposition calculation follows Ghan et al. (2013). ALL indicates the total $\Delta F$ calculated from the difference between CTRL and E1850 (ALL=DIR+IND+RES). IND indicates the $\Delta F$ caused by aerosol-cloud interactions (2nd row, indirect aerosol effect), DIR the $\Delta F$ caused by aerosol-radiation interactions (3rd row, direct aerosol effect), and RES (bottom row) the residual forcing. The clear-sky direct aerosol effect (4th row) is also shown. See section 5.1 for details.

At the surface, the anthropogenic aerosol effect through the interactions with radiation contributes to more than 40% of the total forcing (Figure 13g). Large surface longwave indirect aerosol forcing appears in the NH high-latitude and polar regions (Figure 13f), which is caused by more low liquid cloud induced by anthropogenic aerosols. The thicker liquid clouds can trap more outgoing longwave radiation and emit it back to surface. As is discussed in Section 7, this causes a strong increase in the near surface temperature during the boreal winter season.



**Figure 13.** Geographical distributions of decomposed net (left column), shortwave (middle column), and longwave (right column) $\Delta F$ at the surface. The decomposition calculation follows Ghan et al. (2013). ALL indicates the total $\Delta F$ calculated from the difference between CTRL and E1850 (ALL=DIR+IND+RES). IND indicates the $\Delta F$ caused by aerosol-cloud interactions (2nd row, indirect aerosol effect), DIR the $\Delta F$ caused by aerosol-radiation interactions (3rd row, direct aerosol effect), and RES (bottom row) the residual forcing. The clear-sky direct aerosol effect (4th row) is also shown. See section 5.1 for details.

## 5.2 Decomposition of forcing by individual compositions

To investigate the impact of changes in individual aerosol compositions, we decompose $\Delta F$ based on the different anthropogenic aerosol emission types that contribute to the total forcing. To do this, we change one type of aerosol emissions in E1850 at a time from the pre-industrial condition to 2010 level, so that the contribution of this single emission type can be





estimated. All other aerosol emissions are set to pre-industrial conditions. Note that we take the 1850 aerosol condition as the background. Due to the non-linear effect (e.g., BC mixing state in 1850 is different from that in 2010), the sum of the estimated forcings by individual aerosol species would not be the same as the total forcing estimated using CTRL and E1850.

Figure 14 shows the geographical distributions of the decomposed $\Delta F$ for individual aerosol emission types. The anthropogenic sulfur emission has a large impact on the $\Delta F$ (Figure 14d). The global annual mean net forcing is about $-1.66\,\mathrm{Wm}^{-2}$, which is very close to the $\Delta F$ estimate due to all aerosol emission changes. However, both the shortwave and longwave $\Delta F$ magnitudes increase significantly (by $0.25\,\mathrm{Wm}^{-2}$) and they compensate with each other. The anthropogenic BC emission has an overall positive $\Delta F$ (Figure 14g), with a global annual mean net forcing of $0.27\,\mathrm{Wm}^{-2}$. The positive $\Delta F$ over land
is mainly caused by the shortwave absorption of BC (Figure 14h). The compensating changes over the Tropical Warm Pool region ($\Delta F_{SW}$ and $\Delta F_{LW}$) are mainly caused by reduced high clouds in that area. The impacts of the anthropogenic POM and SOA emissions on $\Delta F$ are overall negative ($-0.3$ to $-0.4\,\mathrm{Wm}^{-2}$, Figure 14j,m). The impacts of biomass burning emission changes (BB) on $\Delta F$ are mostly in the shortwave and they have a distinct regional pattern (Figure 14m,n). Over Siberia, BB $\Delta F$ is positive (mainly in JJA and MAM), while over the west of Columbia, Ecuador, and Peru, there is a significant negative
forcing (in JJA and SON). This suggests there is large differences of BB emissions near (or on the upwind side) these two regions between the pre-industrial (1850) and present-day (2010) emissions.

## 6 Sensitivity of effective aerosol forcing estimation to parameterization changes

Based on the sensitivity simulations performed during the model development, we have identified several parameterization changes that have important impacts on the anthropogenic aerosol effect in the model. These include changes in cloud micro-
physics that affect the auto-conversion in warm clouds, Bergeron process and heterogeneous ice nucleation in mixed-phase clouds, and homogeneous ice nucleation in cirrus clouds (see Table 1).

  As described in Rasch et al. (2019), E3SMv1 uses a modified auto-conversion scheme based on Khairoutdinov and Kogan (2000) (hereafter KK2000), where the parameters have been tuned so that the autoconversion process converts less liquid to rain compared to the standard KK2000 treatment. As a result, the liquid water content increases less rapidly (especially under
the more pristine conditions) when the aerosol and cloud droplet number concentrations are small. With the parameters reset to KK2000 standard values, the liquid water path is decreased significantly in most regions (Figure F1e in Appendix). In contrast, the changes in liquid water path between the PD and PI simulations become larger (Figure 15c and Figure F1f in Appendix). Consequently, the simulated anthropogenic aerosol effect ($-2.03Wm^{-2}$) is much larger than the reference model (Table 2 and Figure 15).
The Bergeron process rate determines how much liquid is converted to ice through the Wegener-Bergeron-Findeisen process. In E3SMv1, a scaling factor was set to make the Bergeron process rate 10% of its nominal value. This was originally intended to compensate for an overestimated ice formation rate when using the Meyers scheme (Meyers et al., 1992). The overly small Bergeron process rate produces too much supercooled liquid clouds in the simulation. By changing the scaling factor from 0.1 to 0.7, the liquid and ice water path is decreased in most places, especially in in mid-latitude and polar regions (Figure F1c and





**Figure 14.** Geographical distributions of the annual mean net (left column), shortwave (middle column), and longwave (right column) effective radiative forcing at TOA by individual aerosol components (i.e., compositions). For example, "E2010SU" indicates the simulation same as E1850 but with present-day (2010) sulfur emissions. See section 5.2 and Table 1 for details.

Figure G1c in Appendix). The changes in liquid and ice water paths between the PD and PI simulations also become smaller (Figure F1d and Figure G1d in Appendix). As a result, the simulated anthropogenic aerosol effect is significantly weaker than the reference model for both the shortwave and longwave components (Table 2 and Figure 15).





**Table 2.** Global and annual mean net, shortwave (SW), and longwave (LW) $ERF_{aer}$ (W m$^{-2}$) at the TOA and surface in the nudged simulations. Model configurations are described in Table 1.

| Model configurations | $\Delta F_{toa}$ | $\Delta F_{toa,SW}$ | $\Delta F_{toa,LW}$ | $\Delta F_{surf}$ | $\Delta F_{surf,SW}$ | $\Delta F_{surf,LW}$ |
|---|---|---|---|---|---|---|
| **Group 2** | | | | | | |
| E2010 (CTRL) - E1850 | -1.64 | -2.43 | 0.79 | -3.09 | -3.81 | 0.73 |
| E1900 - E1850 | -0.38 | -0.52 | 0.14 | -0.46 | -0.61 | 0.15 |
| E1950 - E1850 | -1.03 | -1.47 | 0.45 | -1.36 | -1.73 | 0.36 |
| E1970 - E1850 | -1.66 | -2.41 | 0.75 | -2.25 | -2.81 | 0.55 |
| E2000 - E1850 | -1.69 | -2.49 | 0.80 | -2.74 | -3.39 | 0.65 |
| CMIP5 (E2000 -E1850) | -1.44 | -2.22 | 0.79 | -2.56 | -3.15 | 0.59 |
| **Group 3** | | | | | | |
| E2010SU - E1850 | -1.66 | -2.70 | 1.04 | -2.12 | -2.60 | 0.48 |
| E2010BC - E1850 | 0.27 | 0.39 | -0.12 | -0.38 | -0.55 | 0.17 |
| E2010POM - E1850 | -0.40 | 0.30 | -0.11 | -0.46 | -0.47 | 0.01 |
| E2010SOA - E1850 | -0.31 | -0.30 | -0.01 | -0.40 | -0.45 | 0.05 |
| E2010BB - E1850 | 0.05 | 0.07 | -0.03 | -0.06 | -0.07 | 0.01 |
| **Group 4** | | | | | | |
| BERG07 - BERG07_PI | -1.49 | -2.10 | 0.61 | -2.94 | -3.67 | 0.73 |
| KK2000 - KK2000_PI | -2.03 | -3.02 | 0.99 | -3.46 | -4.42 | 0.96 |
| HOM100 - HOM100_PI | -1.62 | -2.13 | 0.52 | -2.87 | -3.51 | 0.64 |
| MEY - MEY_PI | -1.49 | -2.17 | 0.68 | -3.02 | -3.65 | 0.63 |

E3SMv1 uses the CNT-based heterogeneous ice nucleation scheme to consider the ice formation in mixed phase clouds (Wang et al., 2014). Compared to Meyers scheme (Meyers et al., 1992) used in E3SMv0, the CNT scheme predicts less heterogeneous ice nucleation and freezes less supercooled liquid (Figure 15c and Figure F1h in Appendix), especially in the NH polar region. The increased supercooled liquid amount leads to optically thick liquid clouds in the lower troposphere, which subsequently increases the downward longwave radiation and warm the surface in the NH polar region (this is further explained in the next section). Much larger impact has been seen in the high-resolution simulations (Caldwell et al., 2019), which reported severely overestimated Arctic surface temperature when the CNT scheme was used along with the 0.1 Bergeron process rate scaling factor. This might be related to the fact that the high-resolution model can better resolve the heterogeneity in cloud phase (Tan and Storelvmo, 2016). As mentioned in Caldwell et al. (2019) and above, it is more reasonable to use a larger scaling factor (e.g., 0.7 instead of 0.1) when the parameterization use the CNT scheme. When the Meyers scheme is used, the relative changes in the column-integrated mass and number are reduced for both cloud liquid and ice, leading to a reduced $ERF_{aer}$ in both the shortwave and longwave components. Similar sensitivities were found in the CAM6/CESM2 model simulations (Gettelman et al., 2019).

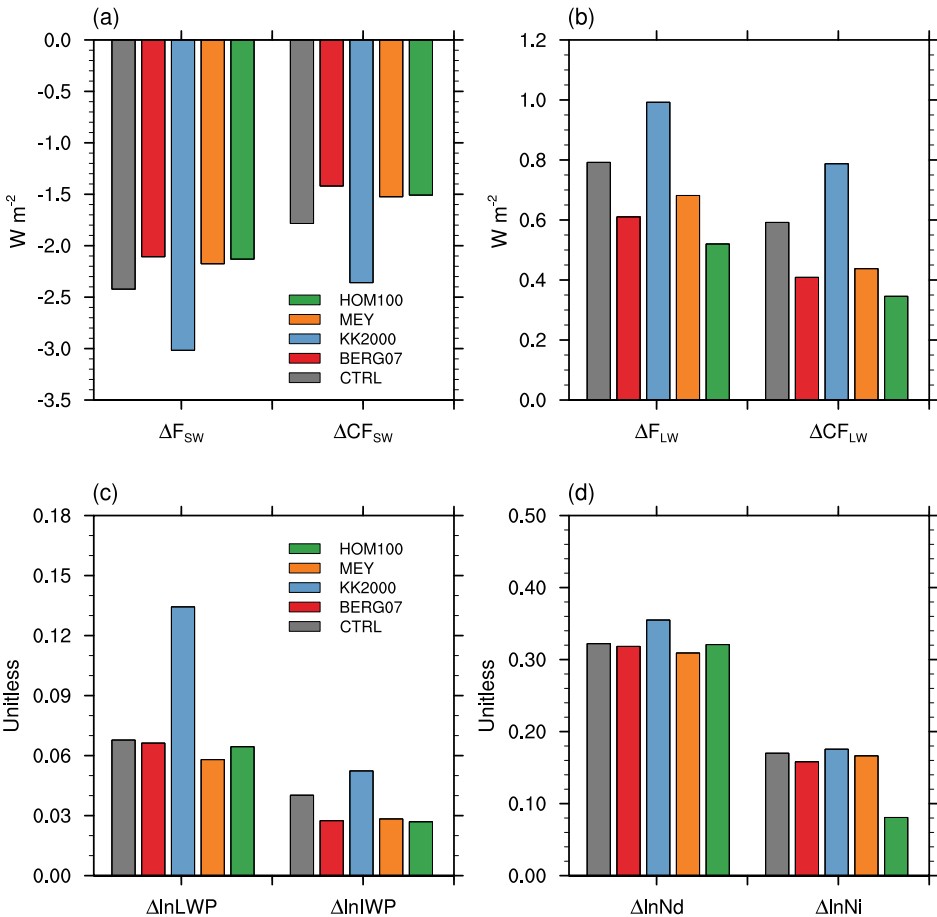

**Figure 15.** Global annual mean anthropogenic aerosol effects on TOA radiative fluxes ($\Delta F$ and $\Delta CF$) and hydrometeor mass/number (relative changes) in the reference and sensitivity simulations. $CF_{SW}$ and $CF_{LW}$ are shortwave and longwave cloud radiative forcings. LWP and IWP are the liquid and ice water path in stratiform clouds. $N_d$ and $N_i$ are the column-integrated droplet and ice number concentrations. $\Delta lnX = \Delta X / X_{PD}$. See section 6 for details.

In E3SMv1, the threshold size of Aitken sulfate aerosols for homogeneous ice nucleation was adjusted to $50\,\text{nm}$ in the DECK simulation setup from the original $100\,\text{nm}$ value used in CAM5. This allows more Aitken mode sulfate particles to nucleate ice homogeneously, which increases the ice crystal number concentration and their subsequent growth greatly. When the threshold size is reset to $100\,\text{nm}$, the ice crystal number and ice water content decrease by about a factor 2 in most regions

(Figure G1i in Appendix) and the changes in ice crystal number and ice water content between the PD and PI simulations become smaller (Figure 15d and Figure G1j in Appendix) . As a result, the simulated anthropogenic aerosol effects in both the shortwave and longwave are reduced compared to the reference model (Table 2). Due to the strong compensation between the changes in SW and LW, the change in net effect is fairly small (-1.62 $\text{Wm}^{-2}$ vs. -1.64 $\text{Wm}^{-2}$ in the reference model).




**Figure 16.** Global annual and seasonal mean changes in near surface temperature at 2m ($\Delta$TREFHT, K), decomposed aerosol indirect effect at the surface ($\Delta F_{\mathrm{IND,SRF}}$, $W\,m^{-2}$), and downward longwave radiation flux ($\Delta$FLDS, W m$^{-2}$) due to the anthropogenic aerosol effects. See section 7 for details.

We note that the sensitivities discussed above are likely dependent on the amount of simulated background (pre-industrial)
aerosols and cloud hydrometeor concentrations in the model. For example, the autoconversion rates estimated by the original and modified KK2000 parameterizations depend on the droplet number concentrations and the difference between them is large under pristine conditions (where the cloud droplet number is small, see Figure 2 in Rasch et al., 2019).

## 7   Impact on near surface temperature over land and ice-covered regions through fast processes

The near surface temperature is strongly affected by the radiative heating/cooling caused directly or indirectly by anthropogenic
aerosols. Since the sea surface temperature is prescribed in our AMIP and nudged simulations, the impact of anthropogenic





aerosols on sea surface temperature and the associated slow processes, such as the ocean circulation and its feedback to atmosphere can not be considered. Nevertheless, it is useful to estimate the impact of anthropogenic aerosols on the surface temperature through fast processes. The impact through slow processes can be inferred, if both the AMIP and coupled simulation results are available.

We first look at the results from the nudged simulations. Since the large-scale circulation and SST are constrained, the local changes in diabatic heating/cooling (affected mainly by fast processes) have a dominant influence on the near surface temperature over land. Figure 16 (first row) shows the near surface temperature (at 2m) changes ($\Delta$TREFHT) caused by the emission changes between pre-industrial and present-day conditions. As expected, near the emission sources and the downwind regions (especially in NH mid-latitudes), anthropogenic aerosols in general have strong cooling effects. However, over the NH

polar region, we see warmer surface temperature associated with responses over sea ice from increased anthropogenic aerosol emissions.

The increase in near surface temperature mainly happens in the boreal winter season (Figure 16b) and in late autumn and early spring, when the shortwave radiation is weak (due to the large solar Zenith angle) and the longwave radiation plays a more important role in determining the near surface temperature. Between simulations using 2010 and 1850 emissions, there

is about 0.5-2K increase in December-January-February mean near surface temperature over land and areas covered by sea ice near the Arctic. During the boreal summer season, the shortwave cooling effect is usually stronger than the longwave warming effect. Also, during summer season the sea ice cover is smaller than in winter, and the near surface temperature is less affected by the surface radiation changes. Therefore, the overall effect is cooling over land areas in the NH polar region (Figure 16c) and over the Arctic Ocean the near surface temperature change is not large.

In order to determine what processes cause the temperature increase in NH polar region, we compared the seasonal mean near surface temperature changes with the decomposed surface anthropogenic aerosol forcing (see previous section). The anthropogenic aerosol indirect effect (through the interactions with clouds) has a strong warming effect at the surface (Figure 16g,h). Furthermore, both the spatial pattern and the seasonal variations agree well with temperature increase. The strong anthropogenic aerosol indirect effect is mainly contributed by the large increase in liquid water path (not shown). The thicker

liquid clouds trap more outgoing longwave radiation and reflect it back to surface ($\Delta$FLDS in Figure 16j,k), which causes the strong temperature increase in boreal winter season. The results are consistent with previous observational studies (Garrett and Zhao, 2006; Lubin and Vogelmann, 2006), which reported significant anthropogenic aerosol indirect effect in the longwave over the Arctic.

Results from the AMIP simulations (Figure 17) show a similar relationship between the changes in near surface temperature

and in the downward longwave radiation due to the anthropogenic aerosol effects. Note that we don't have the decomposed radiative forcing diagnostics output for the AMIP simulation, here only the relationship between $\Delta$TREFHT and $\Delta$FLDS is shown. There is a strong correlation between $\Delta$TREFHT and $\Delta$FLDS time series. The temporal variations in $\Delta$TREFHT and $\Delta$FLDS are mainly affected by the inter-annual variations in both the liquid cloud distributions and the amount of anthropogenic aerosols. The nudged simulations (shown as dots) well capture the changes in both variables. This suggests that



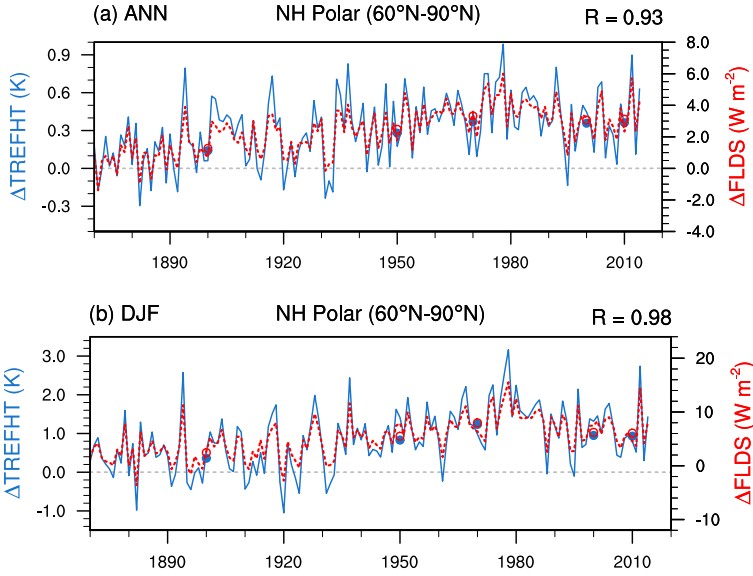

**Figure 17.** Annual (a) and boreal-winter (b) mean changes in near surface temperature at 2m ($\Delta$TREFHT, K, in blue) and downward longwave radiation flux ($\Delta$FLDS, W m$^{-2}$, in red) over the NH polar region due to the anthropogenic aerosol effects. Both the AMIP (lines) and nudged (dots and circles) simulations are shown here. See section 7 for details.

results from the relatively short nudged simulations can provide a reasonable estimate of the simulated near surface temperature responses due to the anthropogenic aerosol forcing through the fast processes.

# 8 Conclusions

This study investigates the historical and present-day ERF$_{aer}$ in E3SMv1 using AMIP simulations for 1870-2014 and short nudged simulations. We analyze the relationship between key aerosol and cloud quantities, decompose the anthropogenic
aerosol forcing in different ways, and evaluate the parameterization sensitivities.

Despite the overall increase in the global mean anthropogenic aerosol burden and optical depth ($\Delta\tau_a$) during 1870-2014, there are large differences in the historical changes of these quantities between different regions and aerosol species. For example, the anthropogenic sulfate burden and sulfate $\Delta\tau_a$ over the NH polar regions and the NH mid-latitude regions decrease significantly after 1970, while for carbonaceous aerosols, there is a continued overall increase in burden and $\Delta\tau_a$. Both the
shortwave and longwave ERF$_{aer}$ estimated by E3SMv1 are outside the two-standard-deviation range of the CMIP6 RFMIP model estimates (Smith et al., 2020), although they are not the strongest among the CMIP6 RFMIP model results. Similar to the changes in anthropogenic aerosol burden and optical depth, we see large regional differences in the temporal evolution of ERF$_{aer}$. The overall increasing or decreasing trend in ERF$_{aer}$ resembles the changes in sulfate $\Delta$burden and $\Delta\tau_a$, especially for NH mid-latitude and polar regions.





The relative changes (dlnY versus dlnX) in historical global annual mean anthropogenic aerosol optical depths, CCN concentrations, and cloud droplet number concentrations show overall linear correlation. However, after around 1970, the correlations show a significant change, mainly caused by the regional differences in the historical changes of anthropogenic aerosol burden and optical depths, as well as their impacts on the CCN and cloud droplet formation. For ice cloud properties, we find similar but weaker correlations compared to those seen in liquid cloud properties.

A forcing decomposition analysis following Ghan et al. (2013) shows that the TOA $\mathrm{ERF_{aer}}$ in EAMv1 is dominated by the anthropogenic aerosol effect through the interactions with clouds. There is a strong compensation between the forcings in the shortwave and longwave components. At the surface, the anthropogenic aerosol effect through the interactions with radiation contributes to more than 40% of the total forcing. Large surface longwave indirect aerosol forcing appears in the NH high-latitude and polar regions, which is caused by the increase of low liquid cloud induced by anthropogenic aerosols. The forcing

analysis decomposed by aerosol species shows that anthropogenic sulfate aerosols have the largest effect (both in the shortwave and longwave) on the overall $\mathrm{ERF_{aer}}$. Anthropogenic BC aerosols exert a strong positive shortwave forcing due to absorption, but a weaker negative longwave forcing by weakening vertical motions and reducing high-cloud amounts. Both anthropogenic POM and SOA have significant negative forcing, but the forcing over the NH high-latitude and polar region is small.

    We have also evaluated the sensitivity of simulated $\mathrm{ERF_{aer}}$ to parameterization changes in EAM. Out of the four changes

that were adopted in EAMv1, resetting the autoconversion tuning parameters (to the original values used in KK2000) has the largest impact on the net forcing (more negative), confirming that the autoconversion tuning parameters applied in EAMv1 indeed helps to reduce the effective aerosol forcing. Two changes are made for mixed-phase clouds, i.e., the tuning of the Bergeron process rate and the change in ice nucleation scheme. Both of them significantly reduce the $\mathrm{ERF_{aer}}$ in the shortwave and longwave components, but the impacts on the net forcing are relatively small. The last change, in homogeneous ice

nucleation, makes the ice formation rate smaller in cirrus clouds. This also significantly reduces the aerosol forcing in both shortwave and longwave components, but the impact on net forcing is negligible.

    The results summarized above suggest that in order to constrain the net $\mathrm{ERF_{aer}}$ in future versions of E3SM, it is important to constrain the relative changes ($\Delta lnX = \Delta X/X$) in cloud droplet number concentrations ($\mathrm{N}_d$) and liquid water path (LWP). In E3SMv1, very small $\mathrm{N}_d$ (e.g., $\mathrm{N}_d < 10\,\mathrm{cm^{-3}}$) appears frequently under pristine conditions.[3] Applying a lower bound of cloud

droplet number concentration can effectively increase $\mathrm{N}_d$ under pristine conditions and reduce $\Delta \ln \mathrm{N}_d$ in some models, though this treatment is artificial and thus not desirable (Hoose et al., 2009). One possible solution to reduce $\Delta \ln \mathrm{N}_d$ in a more physical way is to improve the representation of natural aerosols (especially for sulfate) and the associated cloud droplet formation or depletion processes in the model. The other way is to improve the droplet formation parameterization so that it can include some important processes (e.g., enhanced mixing induced by cloud-top radiative cooling, droplet spectral dispersion, etc.) that

are currently not considered in E3SM, but these might need substantial changes in the model. Finally, as suggested by the sensitivity simulations, increasing the primary and/or secondary ice formation in mixed-phase clouds could also indirectly reduce $\Delta \ln \mathrm{N}_d$ and $\Delta \ln \mathrm{LWP}$.

---

[3]to be presented in detail in a separate study

In response to the aerosol forcing associated with fast processes (in contrast to the slow processes involving ocean/sea ice changes), significant regional changes are seen in the simulated near-surface air temperature. Due to both the strong shortwave

direct and indirect aerosol effects, surface cooling occurs in the mid-latitude and low-latitude regions. Strong surface warming is seen in the NH high-latitude regions during boreal winter, which is mainly caused by denser liquid clouds in these regions caused by the longwave indirect aerosol effect. Since the warming caused by greenhouse gases is stronger over the polar regions than in lower latitudes, the high-latitude warming and mid-/low-latitude cooling caused by anthropogenic aerosols further amplifies the meridional gradient in the near-surface temperature changes. The small change (close to zero) in the

global mean surface temperature in the AMIP and nudged simulations suggests that the surface cooling in the second half of the 20th century simulated by the coupled model is very likely caused by the anthropogenic aerosol effects through slow processes involving ocean/sea ice responses. In the future, we plan to use both the AMIP and coupled simulations to further investigate the anthropogenic aerosol effects on the historical surface temperature change through different processes.

*Code and data availability.* The model results presented in this study and the EAMv1/E3SMv1 source code can be found on Zenodo

(https://doi.org/10.5281/zenodo.5792600 and https://doi.org/10.5281/zenodo.5794575). The EAMv1/E3SMv1 source code can also be found at https://github.com/E3SM-Project/E3SM/tree/v1.0.0 (hash 849d9ee).

*Author contributions.* KZ designed the study, performed (nudged) simulations and analyses, and wrote the first draft of manuscript. WZ contributed to the historical change analysis. HWan contributed to the simulation design and the causality analysis. SJG and MW contributed to the causality and decomposition analysis. PJR, PM, XS, YW, and XL contributed to the configuration and/or interpretation of sensitivity

simulations. RCE, HWang, KZ, PM, SZ, JS, SB, MS, BS, YQ, YF, JHY, SJG, and PJR contributed to the aerosol/cloud model and analysis tool development. JCG, QT, WL, and SX contributed to the configuration and/or simulation of the AMIP runs. All authors contributed to the result discussion and manuscript revisions.

*Competing interests.* Susannah Burrows, Xiaohong Liu, Manish Shrivastava, Hailong Wang, and Yun Qian are Topical Editors of Atmospheric Chemistry and Physics. No competing interests are present for other authors.

**Appendix: Additional Figures**



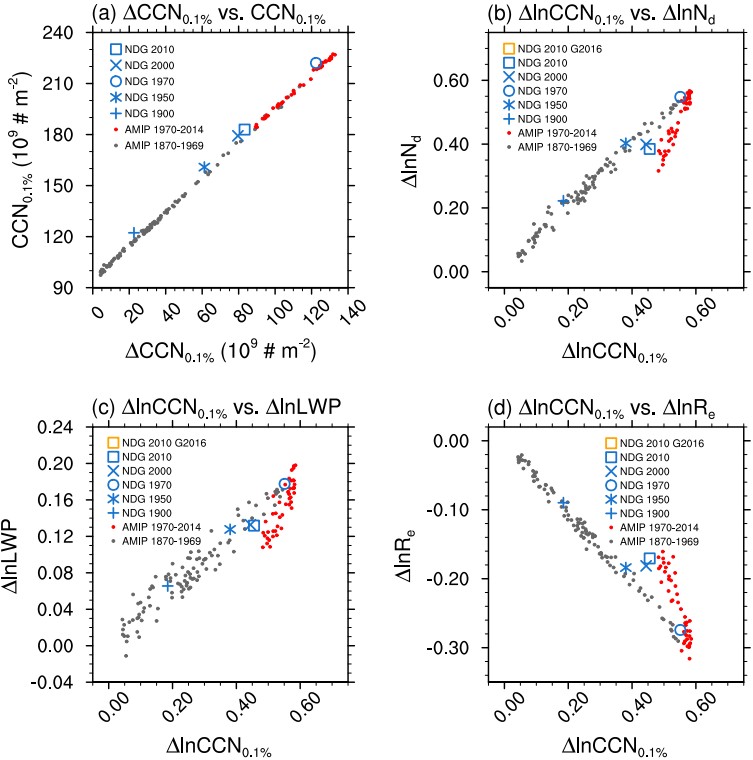

**Figure A1.** As Figure 8, but for the NH polar region. See section 4.2 for details.

*Acknowledgements.* This research was supported as part of the Energy Exascale Earth System Model (E3SM) project (grant no. 65814), funded by the U.S. Department of Energy, Office of Science, Office of Biological and Environmental Research. The authors thank all E3SM team members for their efforts in developing and supporting the E3SM model. The Pacific Northwest National Laboratory (PNNL) is operated for DOE by Battelle Memorial Institute under contract DE-AC06-76RLO 1830. WZ was partly supported by the Visiting Researchers (formerly Alternate Sponsored Fellow) Program at PNNL. YF would like to acknowledge the support of Argonne National Laboratory (ANL) provided by the U.S. DOE Office of Science, under Contract No. DE-AC02-06CH11357. Work at LLNL was performed under the auspices of the US DOE by Lawrence Livermore National Laboratory under contract No. DE-AC52-07NA27344. J. Yoon was supported by the National Research Foundation of Korea. This research used high-performance computing resources from the PNNL Research Computing, and the National Energy Research Scientific Computing Center (NERSC), a DOE Office of Science User Facility supported by the Office of Science of the U.S. Department of Energy under Contract No. DE-AC02-05CH11231.





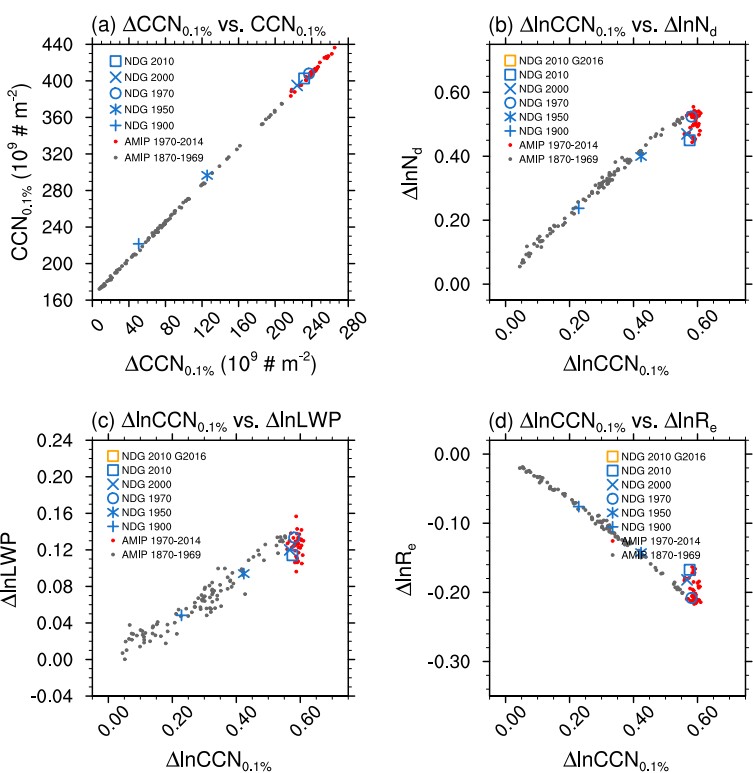

**Figure B1.** As Figure 8, but for the NH mid-latitude region. See section 4.2 for details.





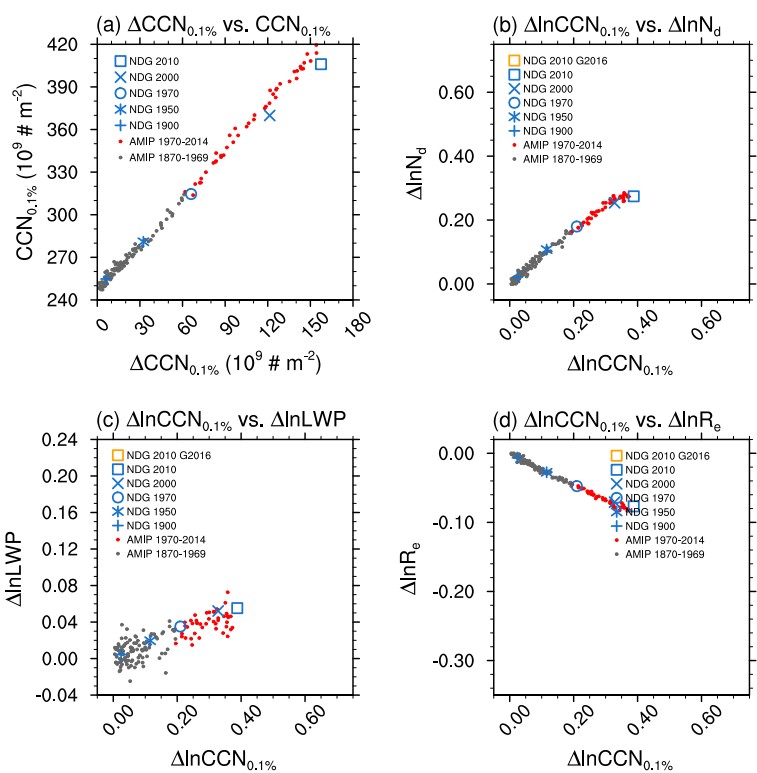

**Figure C1.** As Figure 8, but for the tropics. See section 4.2 for details.



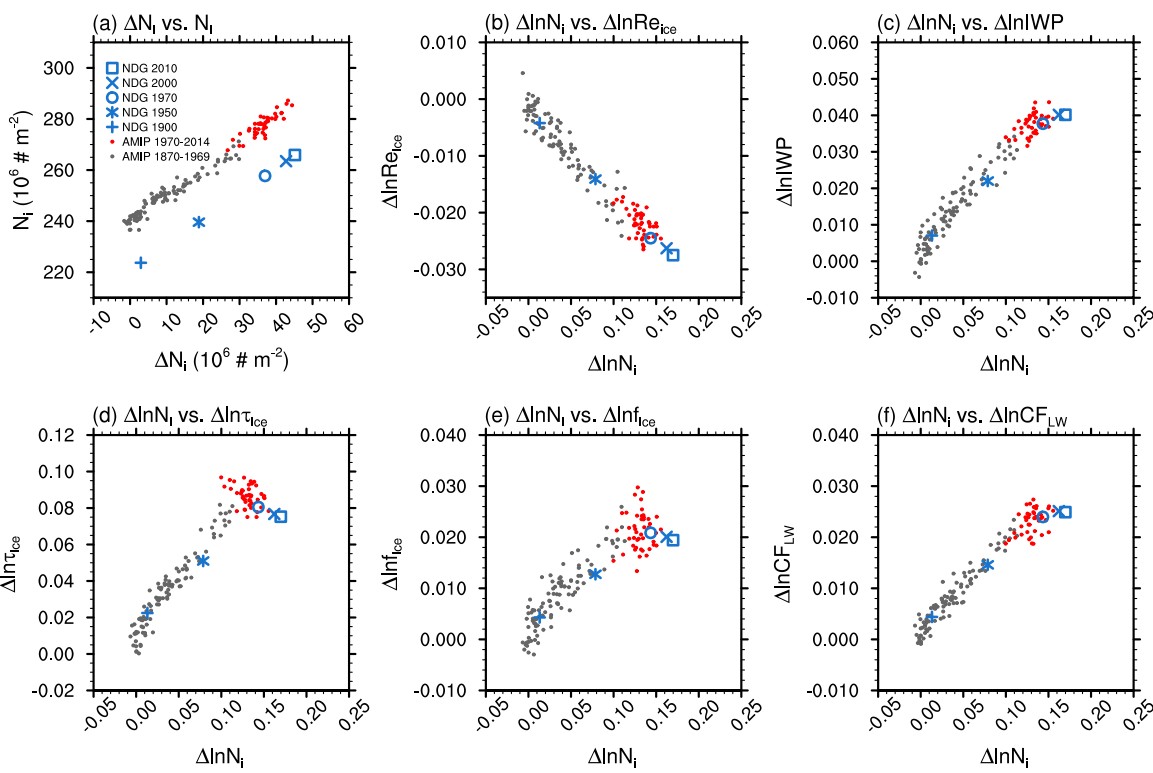

**Figure D1.** As Figure 7, but for relationships between $\Delta N_i$ and $N_i$, and between relative changes in $N_i$ and other quantities. $\Delta lnX = \Delta X/X_{PD}$. See section 4.4 for details.

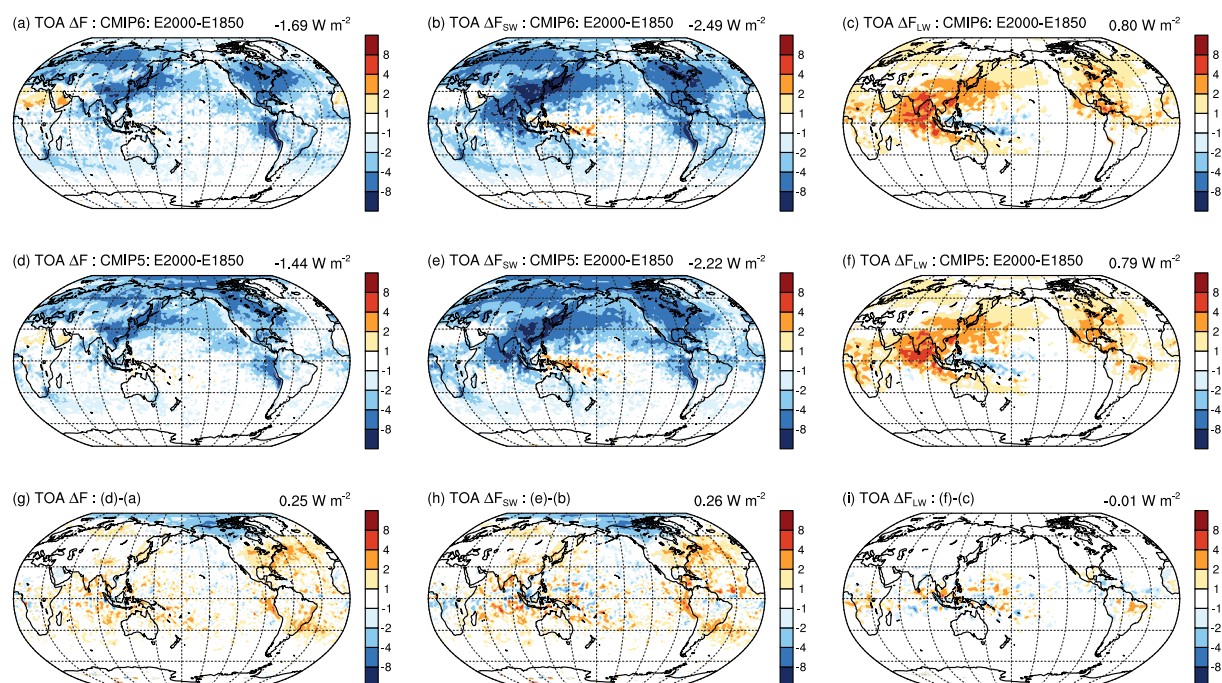

**Figure E1.** Annual mean global distribution of anthropogenic aerosol effects ($\mathrm{W\,m^{-2}}$) estimated with year 2000 CMIP5 (2nd row) and CMIP6 (1st row) emissions.

**Figure F1.** Annual mean global distribution of liquid water path and the sensitivity to aerosol perturbations ($W\,m^{-2}$) in the reference and sensitivity simulations. See section 6 for details.



**Figure G1.** Annual mean global distribution of ice water path and the sensitivity to aerosol perturbations (W m$^{-2}$) in the reference and sensitivity simulations. See section 6 for details.

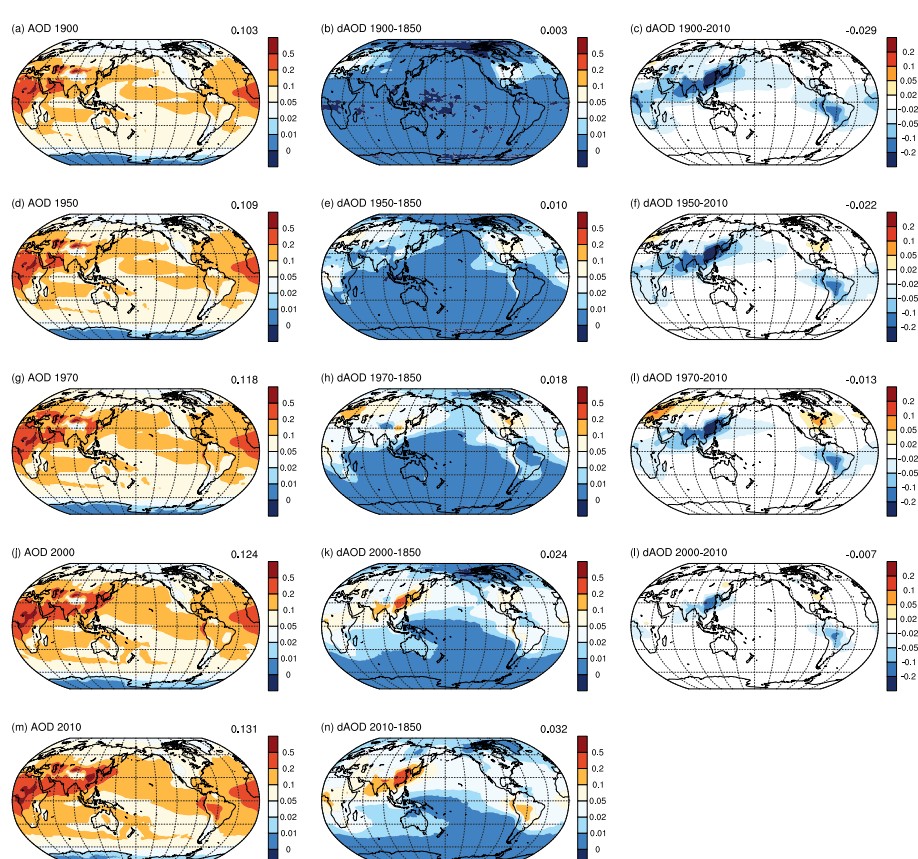

**Figure H1.** Annual mean global distribution of AOD in simulations with emissions for different years. See section 3.2 for details.

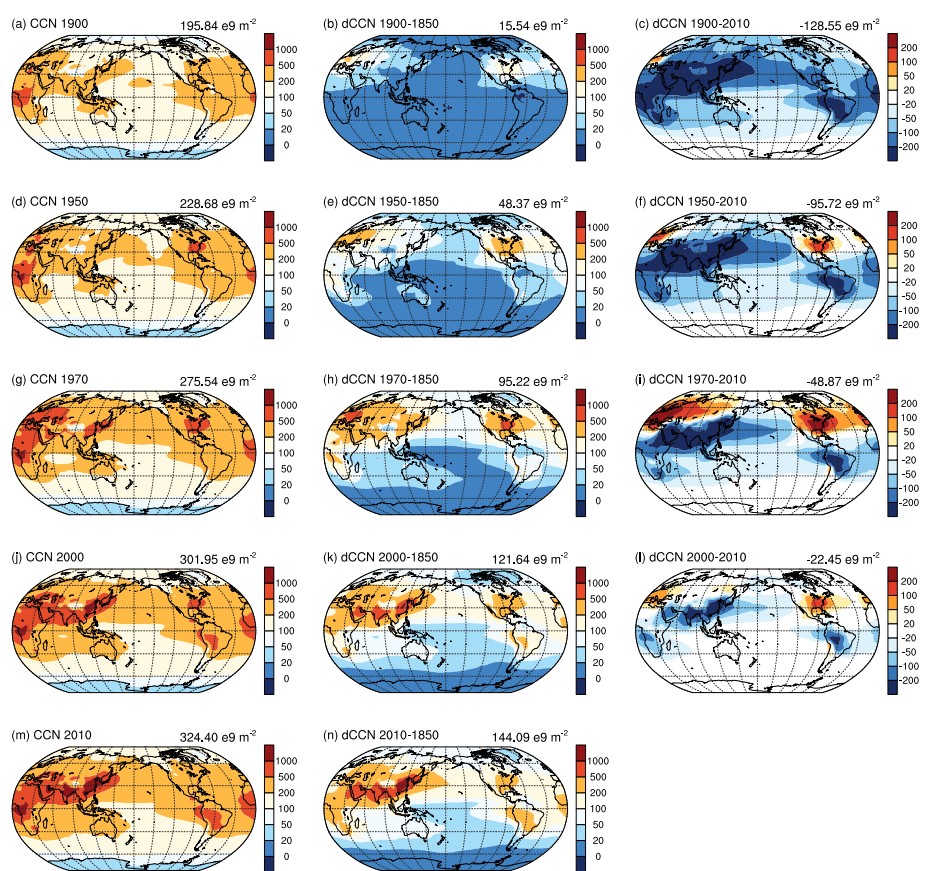

**Figure I1.** Annual mean global distribution of column-integrated CCN (at 0.1% supersaturation) concentrations in simulations with emissions for different years. See section 3.2 for details.



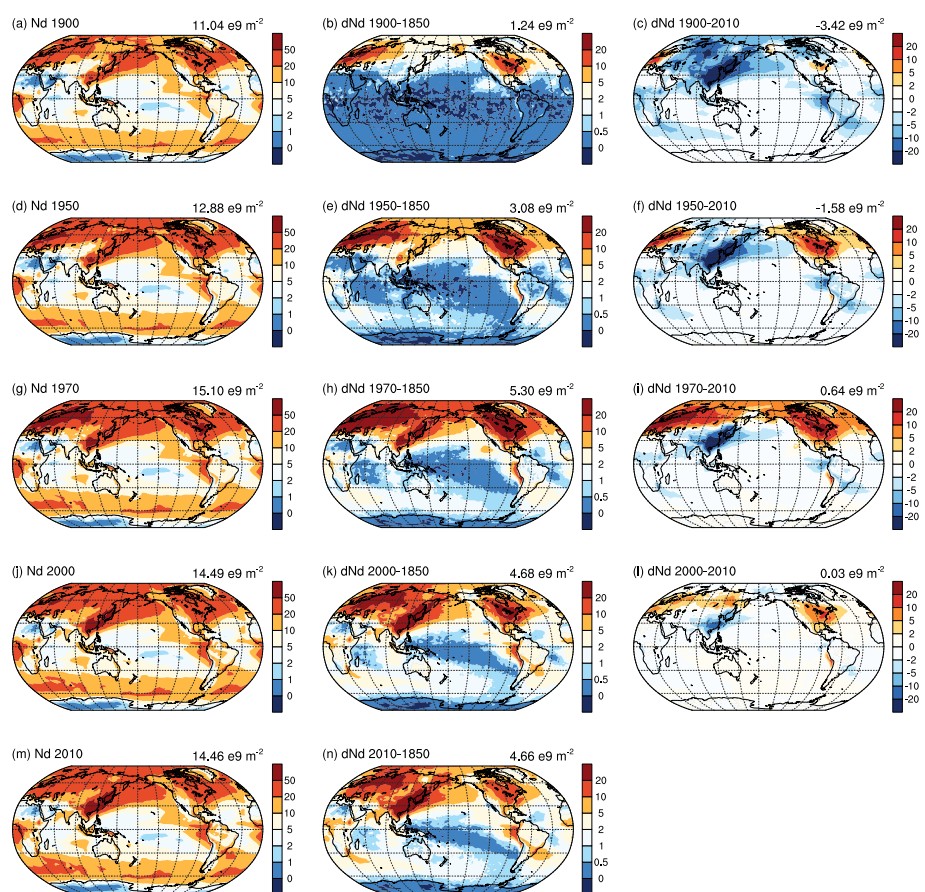

**Figure J1.** Annual mean global distribution of column-integrated cloud droplet number concentrations ($N_d$) in simulations with emissions for different years. See section 3.2 for details.



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
