# Peer review of "Effective radiative forcing of anthropogenic aerosols in E3SMv1: historical changes, causality, decomposition, and parameterization sensitivities"

_Atmospheric Chemistry and Physics, 2021_

## Referee Comment (RC1)

I only have some clarification issues with this paper. It is a straightforward discussion of aerosol impacts in the E3SMv1 model, with only slight insights (i.e. few explanations of why they find the things they find). It would be a better paper if at least some possibilities for why they obtain the results they do, together with graphs/results added to show why they believe these are the causes of their results. I list below areas where clarification is needed.

Line 12-14:  Strange that you say there are linear relations from 1870-2014, while the linear relations diverge after 1970. Do you mean the slope of the linear relationship changes after 1970?

Line 16-17:  you need to explain that the increase in radius is stronger than would be predicted by the increase in LWP, here.

Line 19-20: How much does sulfate affect ice clouds through homogeneous nucleation? This is normally a very small change. Where is this discussed/shown in the manuscript?

Line 121-123: The observed standard deviation of updraft velocity within warm stratiform clouds is 0.5 m/s (Paluch and Lenschow, 1992), so employing a lower bound of 0.2 m/s is too high. What study has been done to justify this as compensation for the potentially underestimated turbulence strength?

Line 140-144: what is the justification for the threshold is used for sulfate aerosols? Is this a tuning decision? If so, this should be stated and explained. In typical parcel model simulations, particles smaller than 50 um can nucleate ice, depending on the updraft velocity.

Line 172: delete "the" in "the emission"

Line 176: Dust and sea salt are also not evaluated in the simulations. What is the reasoning for this choice?

Line 213: change "While" to ", while"

Line 208: here it states that BC is scaled by a factor of 10, but the figure states that it is a factor of 5. Which is correct?

Line 222-223:  you state that anthropogenic sulfate affects the dust life cycle, which seems correct, since it can coat dust, causing more removal by precipitation. However, it seems here that dust decreases when sulfate decreases, which is opposite to my intuition. You casually explain that this is through sulfate causing changes in surface winds and moisture, with no explanation of how or why this occurs. Please add this explanation, and why these indirect effects would be larger than the one I mention above.

Figure 3 caption, line 1: marine organic aerosols are also excluded.

Line 229-231: How do changes in dust cause changes in carbonaceous aerosols? Please restate

Figure 5 cation states " Simplified names (E removed) are used. But E is not removed in any of the figure headings.

Figure 6 and discussion: please explain whether a minimum Nd is applied (and why).

Line 322-323: the inference from figure 7 (comparing 7b and 7c) and from Figure 8b is that Nd decreases in response to increasing CCN after 1970. Does the previous sentence explain this somehow?

Line 354: are the reported $\Delta$lnLWP, $\Delta$lnIWP, and cloud optical depth grid-average values or in-cloud values?

Line 384-386: please explain how sulfate aerosols are calculated. Since the aerosol model only treats sulfate mixed with other species, how is sulfate aerosol calculated such that homogeneous nucleation can occur? Or, is there no homogeneous nucleation of cirrus clouds? Or is the effect of sulfate here mainly the result of sulfate deposition on and mixture with, for example, dust aerosol?

Line 396-399: You might explain that F and $F_{clean}$ both include thee indirect effect, so this difference assumes that the indirect effect is the same in F and $F_{clean}$.

Line 466-471: what is the physical reasoning behind the choice of the threshold size of sulfate aerosols (and how are sulfate aerosol sizes determined?) or is this choice just tuning? What in particular is tuned? Was the SW or LW forcing too high compared to other models otherwise?

Line 541-542: How do BC aerosols weaken vertical motions? How do they reduce high-cloud amounts? Why is this sentence here in the conclusions, even though it is not discussed in the paper?

Line 558-560: There is no improvement of natural aerosol emissions that is likely to improve the calculation of $ERF_{aer}$ as much as improving physical processes (i.e. enhanced mixing at cloud top) especially since the model is unlikely to be able to reproduce the darkening expected when aerosols increase as seen in observations (see Zhang, Zhou, Goren, Feingold, ACP, 2022). It is better to understand why one approach vs the other would be a better next step.

---

## Author Comment (AC1)

**Reply to Reviewer #1**

Thank you for the careful review and helpful comments. Below please find our point-by-point response to specific comments.

*I only have some clarification issues with this paper. It is a straightforward discussion of aerosol impacts in the E3SMv1 model, with only slight insights (i.e. few explanations of why they find the things they find). It would be a better paper if at least some possibilities for why they obtain the results they do, together with graphs/results added to show why they believe these are the causes of their results. I list below areas where clarification is needed.*

*Line 12-14: Strange that you say there are linear relations from 1870-2014, while the linear relations diverge after 1970. Do you mean the slope of the linear relationship changes after 1970?*

Yes. The statement is now revised to:

*"... diverging from the linear relationships exhibited for the period of 1870-1969."*

*Line 16-17: you need to explain that the increase in radius is stronger than would be predicted by the increase in LWP, here.*

Line 16-17 says "Compared to other models, E3SMv1 features a stronger sensitivity of the cloud droplet effective radius to changes in the cloud droplet number concentration". If we understand the comment correctly, the reviewer meant the liquid water path (LWP) adjustment will also change the droplet size/effective radius (Re), so the changes in Re are not solely due to changes in the cloud droplet number concentration (Nd).

The formulation we use to express the chain of processes (from changes in aerosols to those in cloud optical properties) allows the feedback and interactions such that changes in Nd affect both Re and LWP and Re is also affected by LWP changes. As pointed out by Ghan et al. (2016), the dlnY/dlnX terms should not be interpreted as only the response of the numerator to changes in the denominator. To avoid confusion, we have revised the sentence to:

*"Compared to other models, E3SMv1 features large relative changes in the cloud droplet effective radius in response to aerosol perturbations."*

*Line 19-20: How much does sulfate affect ice clouds through homogeneous nucleation? This is normally a very small change. Where is this discussed/shown in the manuscript?*

The impact of anthropogenic sulfate on ice clouds through homogeneous ice nucleation is strong in E3SMv1. This is discussed in section 6 of the original manuscript (page 28). When the threshold size of Aitken sulfate aerosols for homogeneous ice nucleation is increased from 50 nm to 100 nm, we see a strong reduction in the ice crystal number and ice water content for both present-day (PD) simulation and the difference between PD and PI. The impact on the net effective aerosol forcing is small, because there is a strong compensation between the LW and

SW forcing changes. In the revised manuscript, we have added the following figure (R1.1) in the appendix to further show the PD-PI changes in simulated ice crystal number vertically integrated above 300hPa, where most ice crystals are formed through the homogenous ice nucleation.

[Figure]

Figure R1.1 Similar as Figure J1, but annual mean global distribution of ice crystal number concentrations (Ni) vertically integrated above 300hPa in simulations with emissions for different years.

*Line 121-123: The observed standard deviation of updraft velocity within warm stratiform clouds is 0.5 m/s (Paluch and Lenschow, 1992), so employing a lower bound of 0.2 m/s is too high. What study has been done to justify this as compensation for the potentially underestimated turbulence strength?*

E3SMv1 inherited this lower bound of 0.2 m/s (for the characteristic updraft velocity) from the CESM (CAM5.4) model. We are not aware of any specific study on justifying the application of this lower bound in CESM/E3SM. We agree that the 0.2 m/s lower bound might be too high to be

applied to all conditions but removing it in E3SMv1 will degrade the cloud simulation if the model is not further tuned. In Ma et al. (2022) and Golaz et al. (2022), this lower bound is reduced to 0.1 m/s, in combination with other tunings in the cloud and turbulence parameterizations. We have added some discussion on this in section 2.1.

Reference

*Ma, P.-L., Harrop, B. E., Larson, V. E., Neale, R. B., Gettelman, A., Morrison, H., Wang, H., Zhang, K., Klein, S. A., Zelinka, M. D., Zhang, Y., Qian, Y., Yoon, J.-H., Jones, C. R., Huang, M., Tai, S.-L., Singh, B., Bogenschutz, P. A., Zheng, X., Lin, W., Quaas, J., Chepfer, H., Brunke, M. A., Zeng, X., Mülmenstädt, J., Hagos, S., Zhang, Z., Song, H., Liu, X., Pritchard, M. S., Wan, H., Wang, J., Tang, Q., Caldwell, P. M., Fan, J., Berg, L. K., Fast, J. D., Taylor, M. A., Golaz, J.-C., Xie, S., Rasch, P. J., and Leung, L. R.: Better calibration of cloud parameterizations and subgrid effects increases the fidelity of the E3SM Atmosphere Model version 1, Geosci. Model Dev., 15, 2881–2916, https://doi.org/10.5194/gmd-15-2881-2022, 2022.*

*Golaz, J.-C., Van Roekel, L. P., Zheng, X., Roberts, A., Wolfe, J. D., Lin, W., Bradley, A., Tang, Q., Maltrud, M. E., Forsyth, R. M., and et al.: The DOE E3SM Model Version 2: Overview of the physical model, Earth and Space Science Open Archive, p. 61, https://doi.org/10.1002/essoar.10511174.1, 2022.*

*Line 140-144: what is the justification for the threshold used for sulfate aerosols? Is this a tuning decision? If so, this should be stated and explained. In typical parcel model simulations, particles smaller than 50 um can nucleate ice, depending on the updraft velocity.*

Yes, it is a tunable parameter in E3SM. The size threshold was first introduced in the CAM5 model to better reproduce observations (Neale et al., 2010, page 135). E3SMv1 inherited this treatment and further tuned the threshold mainly based on the evaluation of cloud radiative forcing. Without this threshold, the simulated high cloud fraction and cloud radiative forcing magnitude are greatly overestimated compared to the satellite retrievals.

We have adjusted the description in the revised manuscript:

*"Sulfate aerosols (or sulfate solution droplets) in the Aitken mode with diameter larger than a threshold are considered as ice-nucleating aerosols for the homogenous ice nucleation. This size threshold was first introduced in the CAM5 model to better reproduce observations (Neale et al., 2010, page 135) and it was set differently in various modeling studies. For example, a threshold size of 100 nm was used in the CAM5 model, while in some sensitivity studies (e.g., Liu et al., 2012b; Zhang et al., 2013; Shi et al., 2015), all Aitken mode particles are considered as potential ice-nucleating aerosols in cirrus clouds. E3SMv1 inherited this treatment and further tuned the threshold mainly based on the evaluation of cloud fraction and cloud radiative forcing."*

Reference:

*Neale, R.B., Chen, C.C., Gettelman, A., Lauritzen, P.H., Park, S., Williamson, D.L., Conley, A.J., Garcia, R., Kinnison, D., Lamarque, J.F. and Marsh, D., 2010. Description of the NCAR community atmosphere model (CAM 5.0). NCAR Tech. Note NCAR/TN-486+ STR, 1(1), pp.1-12.*

*Line 172: delete "the" in "the emission"*

Done.

*Line 176: Dust and sea salt are also not evaluated in the simulations. What is the reasoning for this choice?*

We focus on the effective radiative forcing of anthropogenic aerosols. DMS is a natural aerosol precursor, but in our AMIP historical simulations it is prescribed with different values for 1850 and the present-day condition. This has been explained in the original manuscript. Dust and sea salt emissions are calculated online/interactively in the model and are not considered a forcing agent, but rather, are part of the natural variability and feedback of the perturbed climate and Earth System. Additionally, dust and sea salt burdens do not change dramatically in our model during the historical period (as shown in Figure 1 of the original manuscript). Consequently, they contribute to historical responses of the model primarily through their role in providing part of the natural background aerosol population, whereas in this paper we focus primarily on the model's response to the strong forcings associated with anthropogenic aerosols. Nevertheless, we briefly discussed the burden and aerosol optical depth changes of these two aerosol species.

The following text is added in the revised manuscript:

*"Dust, sea salt, and marine organic aerosols are not considered as a forcing agent, since their emissions are calculated online/interactively in the model and are mainly affected by the natural variability and feedback of the perturbed climate and Earth System."*

*Line 213: change "While" to ", while"*

Done.

*Line 208: here it states that BC is scaled by a factor of 10, but the figure states that it is a factor of 5. Which is correct?*

It should be 5 (instead of 10). Corrected.

*Line 222-223: you state that anthropogenic sulfate affects the dust life cycle, which seems correct, since it can coat dust, causing more removal by precipitation. However, it seems here that dust decreases when sulfate decreases, which is opposite to my intuition. You casually explain that this is through sulfate causing changes in surface winds and moisture, with no explanation of how or why this occurs. Please add this explanation, and why these indirect effects would be larger than the one I mentioned above.*

Our original statement was based on the dust mass budget analysis using data from the nudged simulations. In the nudged simulations, we only weakly constrain the large-scale horizontal

winds, so the near surface winds can still be affected by stability changes in the lower troposphere. Compared to 1850, the global mean dust emission rate decreased by 2-3% in the years after 1970. We didn't see increases in the dust wet removal rate. Instead, to balance the decrease in dust emission (source), both wet and dry removal rates decreased after 1970. In our single forcing sensitivity tests (changing emissions one at a time for individual aerosol species), we find a small reduction in dust emission in all simulations when individual anthropogenic aerosol emissions are changed to present-day conditions. The slightly weakened dust emission and surface wind speeds are very likely due to the changed atmospheric stability in the boundary layer caused by anthropogenic aerosols. Previous studies (e.g., Jacobson and Kaufman 2006, Baro et al., 2017) have reported that aerosols can affect surface winds. We have added the above discussions in the revised manuscript.

We have added the following discussions in the revised manuscript:

*"To understand what causes the decreasing trend in the dust aerosol burden, the dust mass budget is evaluated using the nudged simulations (the impact of inter-annual variability in the AMIP simulations can be avoided). We find the small decreasing trend is mainly caused by slightly weakened dust emission in simulations with increased anthropogenic emissions. Compared to the simulation with 1850 emissions, the global mean dust emission rate decreased by 2-3% in the simulations with emissions after 1970. In the nudged simulations, we only weakly constrain the large-scale horizontal winds, so the near surface winds can still be affected by stability changes in the lower troposphere. The slightly weakened dust emission is very likely due to the changed surface winds caused by anthropogenic aerosols through changes in the atmospheric stability in the lower troposphere (Jacobson and Kaufman 2006, Baro et al., 2017). The dust wet removal rate also decreases with increased anthropogenic emissions, suggesting the impact of anthropogenic aerosols on dust wet scavenging (through coating) is less important in our model compared to the dust emission changes."*

References:

*Jacobson, M. Z., and Kaufman, Y. J. (2006), Wind reduction by aerosol particles, Geophys. Res. Lett., 33, L24814, doi:10.1029/2006GL027838.*

*Baró, R., Lorente-Plazas, R., Montávez, J. P., and Jiménez-Guerrero, P. (2017), Biomass burning aerosol impact on surface winds during the 2010 Russian heat wave, Geophys. Res. Lett., 44, 1088– 1094, doi:10.1002/2016GL071484.*

*Figure 3 caption, line 1: marine organic aerosols are also excluded.*

Now it is mentioned in the figure caption.

*Line 229-231: How do changes in dust cause changes in carbonaceous aerosols? Please restate*

We did not mean to imply that. The sentence has been revised to:

*"The inter-annual variability in total Δτₐ is mainly caused by changes in dust aerosols and by changes in carbonaceous aerosols (BC, POM, SOA, affected by the variation in biomass burning emissions)."*

*"The inter-annual variability in total $\Delta\tau_a$ is mainly caused by changes in dust aerosols and by changes in carbonaceous aerosols (BC, POM, SOA, affected by the variation in biomass burning emissions)."*

*Figure 5 cation states " Simplified names (E removed) are used. But E is not removed in any of the figure headings.*

Thanks for pointing out this. The caption has been corrected.

*Figure 6 and discussion: please explain whether a minimum Nd is applied (and why).*

No minimum Nd is applied in E3SMv1, but a lower bound for the characteristic updraft velocity is applied to compensate for the potentially underestimated turbulence strength. This is now explicitly mentioned in section 2.1 (model description) and section 3 in the revised manuscript.

*Line 322-323: the inference from figure 7 (comparing 7b and 7c) and from Figure 8b is that Nd decreases in response to increasing CCN after 1970. Does the previous sentence explain this somehow?*

The previous sentence is

*"This is likely related to the fact that there is a continuous increase of anthropogenic carbonaceous aerosols in most regions (that increases $\tau_a$) but decreased or stabilized anthropogenic sulfate aerosols (that decreases Nd) in the NH high- and mid-latitude regions."*

We think this is one reason why global annual mean Nd decreases in response to increasing CCN.

To further address this comment, we show the regional mean dlnNd and dlnCCN values from the nudged simulations in the Figure R1.2 below. The large regional differences in dlnNd and dlnCCN contribute to the temporal variations of global mean values. For example, dlnCCN increases continuously in the tropics and is stabilized after 1970 in the mid-latitude (as a result of the increase in Asia and the decrease in North America). This leads to an increase in global mean dlnCCN. In contrast, the global mean dlnNd is greatly affected by the decreases in both the NH polar and NH mid-latitude regions, so it decreases slightly after 1970. Although dlnNd still increases continuously in the tropics, the value is much smaller than those in the NH polar and NH mid-latitude regions. The impact of carbonaceous aerosol increase is most evidently seen from the differences between 2000 and 2010. dlnCCN slightly increases in the NH polar and NH mid-latitude regions, while dlnNd decreases in both regions. An increase in carbonaceous aerosol mass reduces the bulk hygroscopicity of particles and increases the critical supersaturation for aerosol activation. Please also see our reply to reviewer #2 about the "saturation effect", which is relevant to this comment as well.

We have added the above discussion in the revised manuscript.

[Figure]

Figure R1.2: Global and regional annual mean relative changes in CCN (dlnCCN) and Nd (dlnNd) due to differences in anthropogenic aerosols between PI (1850) and PD (2010).

*Line 354: are the reported DlnLWP, DlnIWP, and cloud optical depth grid-average values or in-cloud values?*

For the sampling using monthly model output, DlnLWP and DlnIWP (for stratiform clouds) are calculated using grid-mean values. Cloud optical depth data are in-cloud values, which are sampled using the COSP simulator and derived during post-processing.

For the sampling using high-frequency data following Ghan et al. (2016), these are in-cloud values calculated using the grid-box mean values divided by the cloud cover after averaging in time and space (following the AeroCom protocol https://wiki.met.no/aerocom/indirect). Because they are relative changes in global annual mean quantities, using in-cloud values (derived from grid-mean values and cloud cover after averaging in time and space) or grid-average values give very similar results.

We have clarified this in the revised manuscript.

*Line 384-386: please explain how sulfate aerosols are calculated. Since the aerosol model only treats sulfate mixed with other species, how is sulfate aerosol calculated such that homogeneous nucleation can occur? Or, is there no homogeneous nucleation of cirrus clouds? Or is the effect of sulfate here mainly the result of sulfate deposition on and mixture with, for example, dust aerosol?*

Our model does include the homogeneous ice nucleation in cirrus clouds based on Liu and Penner (2005) and Gettelman et al. (2010). We use the aerosol number concentration in the Aitken mode truncated at a specified cutoff diameter (50 nm in CTRL) to estimate the number of sulfate aerosol

particles that can initiate homogeneous ice nucleation. This is similar to the treatment in the CESM model, but the cutoff size is different.

We have added the following explanation in the revised manuscript:

*"In contrast, the largest changes in Ni are mainly in the tropics and sub-tropics regions, where the cirrus clouds are affected by anthropogenic sulfate particles through the homogeneous ice nucleation. In our model, the homogeneous ice nucleation in cirrus clouds is parameterized based on Liu and Penner (2005) and Gettelman et al. (2010). We use the aerosol number concentration in the Aitken mode truncated at a specified cutoff diameter (50 nm in CTRL) to estimate the number of sulfate aerosol particles that can initiate homogeneous ice nucleation. Similar to findings in Gettelman et al. (2010), the parameterized homogeneous ice nucleation is sensitive to the sulfate aerosol number and there is a large difference between PD and PI simulations (this is further discussed in section 6)."*

Reference:

*Liu, X. and Penner, J.E., 2005. Ice nucleation parameterization for global models. Meteorologische Zeitschrift, pp.499-514.*

*Gettelman, A., Liu, X., Ghan, S. J., Morrison, H., Park, S., Conley, A. J., Klein, S. A., Boyle, J., Mitchell, D. L., and Li, J.-L. F. (2010), Global simulations of ice nucleation and ice supersaturation with an improved cloud scheme in the Community Atmosphere Model, J. Geophys. Res., 115, D18216, doi:10.1029/2009JD013797.*

*Line 396-399: You might explain that F and Fclean both include the indirect effect, so this difference assumes that the indirect effect is the same in F and Fclean.*

Thanks for your suggestion. We have added the following explanations in the revised manuscript:

*"We note that F and $F_{clean}$ for a certain condition (e.g., present-day) are calculated in a single simulation. They both include the impact of indirect aerosol effect, which is considered the same for the two terms in the decomposition calculation."*

*Line 466-471: what is the physical reasoning behind the choice of the threshold size of sulfate aerosols (and how are sulfate aerosol sizes determined?) or is this choice just tuning? What in particular is tuned? Was the SW or LW forcing too high compared to other models otherwise?*

The threshold size was chosen/tuned mainly based on observational constraints (e.g., simulated longwave and shortwave cloud radiative effects and ice crystal number concentration). Yes, if no or very small threshold size is set, the cloud radiative effect will be too strong and ice crystal number concentrations will be greatly overestimated. Conversely, if a large threshold size is chosen, the cloud radiative effect (especially for the longwave) could be too weak.

*Line 541-542: How do BC aerosols weaken vertical motions? How do they reduce high-cloud amounts? Why is this sentence here in the conclusions, even though it is not discussed in the paper?*

The original discussion was removed in the results section, but we forgot to change the conclusion. Now this sentence is removed in the revised manuscript.

We did observe a clear reduction in ice water path and high-cloud amount near the Pacific warm pool region for the effective radiative forcing of BC, which leads to a negative LW indirect aerosol effect (cooling). In our model, BC and POM don't nucleate ice and additional homogenous ice nucleation of sulfate aerosols will increase the high-cloud amounts. Therefore, the reduced ice water path and high-cloud amount are very likely caused by strong direct and semi-direct effects of anthropogenic BC aerosols, which might weaken vertical motion. Testing this hypothesis would require additional sensitivity simulations (e.g., switching off the BC aerosol effect on radiation in the model).

*Line 558-560: There is no improvement of natural aerosol emissions that is likely to improve the calculation of ERFaer as much as improving physical processes (i.e. enhanced mixing at cloud top) especially since the model is unlikely to be able to reproduce the darkening expected when aerosols increase as seen in observations (see Zhang, Zhou, Goren, Feingold, ACP, 2022). It is better to understand why one approach vs the other would be a better next step.*

Our suggestion that ERFaer could be improved in E3SM through improving the natural aerosol emissions is mainly due to the fact that the cloud droplet number concentrations in pristine regions were found to be too small (e.g., $< 10$ cm$^{-3}$) and this could potentially be caused by the emissions and production of natural aerosol in these regions being too low. Further, our sensitivity simulations show that adding a lower bound on cloud droplet number concentrations can help to reduce ERFaer. However, this is only a hypothesis and it is not clear whether improving the natural aerosol representation can reduce the aerosol effect in a physical way or not. It is not our intent to indicate this approach is better than the other ones listed in the conclusion.

Thanks for the reference. We agree that it's hard for a GCM like E3SMv1 to realistically simulate the enhanced mixing at the cloud top and the subsequent changes in cloud properties.

We have revised the conclusions to clarify this:

*"Previous modeling studies have shown that the simulated aerosol radiative forcing is sensitive to the DMS emission (Carslaw et al., 2013) and oxidation (Fung et al., 2022) treatments. The other way is to improve the droplet formation parameterization so that it can include some important processes (e.g., enhanced mixing induced by cloud-top radiative cooling, droplet spectral dispersion, etc.). These processes usually can only be resolved in model simulations at very high-resolution (e.g., Zhang et al., 2022). How to parameterize them accurately in global models will be a challenge and substantial changes might be needed for models like E3SM."*

Reference:

Carslaw, K.S., Lee, L.A., Reddington, C.L., Pringle, K.J., Rap, A., Forster, P.M., Mann, G.W., Spracklen, D.V., Woodhouse, M.T., Regayre, L.A. and Pierce, J.R., 2013. Large contribution of natural aerosols to uncertainty in indirect forcing. Nature, 503(7474), pp.67-71.

Fung, K. M., Heald, C. L., Kroll, J. H., Wang, S., Jo, D. S., Gettelman, A., Lu, Z., Liu, X., Zaveri, R. A., Apel, E. C., Blake, D. R., Jimenez, J.-L., Campuzano-Jost, P., Veres, P. R., Bates, T. S., Shilling, J. E., and Zawadowicz, M.: Exploring dimethyl sulfide (DMS) oxidation and implications for global aerosol radiative forcing, Atmos. Chem. Phys., 22, 1549–1573, https://doi.org/10.5194/acp-22-1549-2022, 2022.

---

## Author Comment (AC2)

**Reply to reviewer #2**

Thank you for the positive feedback and the helpful comments. Below please find our point-by-point response to specific comments.

*In this study, the authors make an exhaustive analysis of the effective radiative forcing of aerosols (ERFaer) simulated by the E3SMv1 climate model. The authors find that anomalies in aerosol amounts and optical depth follow the prescribed emission trends, but that cloud responses are more complex, exhibiting a change in behaviour from the 1970s. That change is traced back to a change in aerosol composition and different regional trends. The study also highlights a sizeable contribution to ERFaer of the longwave part of the electromagnetic spectrum, and the strong dependence of ERFaer to a few parameters. That authors can convincingly explain the mechanisms of that dependence.*

*The paper is very well written. The figures are of high quality and illustrate the discussion very well. The paper is especially interesting because aerosol representations in E3SMv1 are more complex than in most climate models. Although it is sobering that this complexity is still at the mercy of a few parameters, as discussed in section 6, it also means that the authors can perform a detailed process-based analysis.*

*I only have a few comments, mostly aimed at clarifying the discussion in places. For that reason, I recommend publication after minor revisions.*

**Main comments:**

*The authors make a convincing case that the change in the response of cloud microphysics to aerosol perturbations after 1970 is due to a shift from sulfate to carbonaceous in the anthropogenic aerosol composition. I was also expecting a contribution from "saturation effects" due to the non-linear nature of aerosol-cloud interactions (aci), as argued for example by Stevens (2015 https://doi.org/10.1175/JCLI-D-14-00656.1). Some regions could have reached saturation of their aci, and hence throwing more aerosols at clouds does not exert a radiative forcing anymore. Is there no saturation effect in the model? Or is that effect seen on Figure 8d and discussed as regional effects in lines 345-347?*

Thanks for the comment and reference. It appears to us that the CCN-Nd relationship in E3SMv1 has a "saturation effect". As shown in the figure below (3rd row), the slope (dlnNd/dlnCCN) calculated using regional mean fields slightly decreases from 1900 to 2010 for all the inspected regions. This indicates the susceptibility of cloud droplet number concentration to CCN decreases when there are more anthropogenic aerosols. The large regional differences in dlnNd and dlnCCN contribute to the temporal variations of global mean values. For example, dlnCCN increases continuously in the tropics and is stabilized after 1970 in the mid-latitude (because of the increase in Asia and the decrease in North America). This leads to an increase in global mean dlnCCN. In contrast, the global mean dlnNd is greatly affected by the decreases in both the NH polar and NH mid-latitude regions, so it decreases slightly after 1970. Although dlnNd still increases continuously in the tropics, the value is much smaller that those in the NH polar and NH midlatitude regions. The impact of carbonaceous aerosol increase is most evidently seen from the differences between 2000 and 2010. dlnCCN slightly increases in the NH polar and NH mid-latitude regions, while dlnNd decreases in both regions. The increase in carbonaceous aerosol mass reduces the bulk hygroscopicity of particles and increases the critical supersaturation for aerosol activation.

We have added the following figure (R2.1) and the discussion above in the revised manuscript and more explicitly mentioned the saturation effect as one of the reasons for the change in cloud microphysics response in the conclusion:

*"The relative changes (dlnY versus dlnX) in historical global annual mean anthropogenic aerosol optical depths, CCN concentrations, and cloud droplet number concentrations show overall linear correlation. However, after around 1970, the correlations show a significant change. This is mainly caused by 1) the regional differences in the historical changes of anthropogenic aerosol burden and optical depths, as well as their impacts on the CCN and cloud droplet formation; 2) a shift from sulfate to carbonaceous aerosols in the anthropogenic aerosol composition; and 3) the "saturation effect", as the slope dlnNd/dlnCCN slightly decreases from 1900 to 2010 for all the inspected regions."*

[Figure]

Figure R2.1: Global and regional annual mean relative changes in CCN (dlnCCN, top row) and Nd (dlnNd, middle row) due to differences in anthropogenic aerosols between PI (1850) and PD (2010). The bottom row shows the ratio between dlnNd and dlnCCN.

*After reading the paper, I was left unclear about the source of the longwave component of ERFaer. Is that due mostly to liquid cloud adjustments, or to the ice cloud response?*

It is mainly due to the ice cloud response through homogeneous ice nucleation. This is discussed briefly in section 6. Since both reviewers suggested that this is an important point to discuss, we have revised the abstract, section 4 and 6, and conclusion to emphasize this.

**Other comments:**

*Line 24: "to reduce the magnitude of the net ERFaer" comes as a surprise because the previous paragraph does not explicitly say that the simulated ERFaer is too strong. I suggest clarifying the conclusions of the previous paragraph, perhaps based on lines 50-52.*

We have added one sentence at the beginning of the current paragraph:

*"As suggested by Golaz et al. (2019), the large ERFaer appears to be one of the reasons why the model cannot reproduce the observed global mean temperature evolution in the second half of the 20th century. Therefore, sensitivity simulations are performed to understand which parameterization/parameter changes have a large impact on the simulated ERFaer."*

*Line 33: Could update the references to Chapters 6 and 7 of the AR6 here.*

Thanks for the suggestion. Done.

*Line 49: "is expected to be larger". Is it? All the complex interactions do not necessarily exert radiative forcings of the same sign.*

We have changed it to *"might be larger"*.

*Lines 94-95: It would be useful to summarise here the conclusions of those evaluations of simulated clouds, because they are relevant to aerosol-cloud interactions. For example, Zhang et al. (2019) says in its abstract "generally underestimate clouds in low latitudes and midlatitudes". Does that have implications for the radiative forcing of aerosol-cloud interactions?*

Thanks for the suggestion. We have now included a brief summary about the model evaluations of cloud simulation.

*Lines 121 and 139: Has someone looked at the sensitivity of ERFaer to that lower bound of updraft velocity for liquid and ice nucleation in E3SM? Back in the 2000s, Corinna Hoose and Trude Storelvmo have shown that the use of lower thresholds invites caution. Ok, it is mentioned in the conclusion at line 556, but it could be worth mentioning that issue here too.*

Yes, the sensitivity of applying different lower bounds for droplet and ice nucleation has been tested before, although slightly different model configurations were used. Following the reviewer's suggestion, we have now briefly discussed this issue in section 2.1.2.

*Line 150: To clarify, which version of the CEDS emissions is used? There have made sizeable revisions to sulfur dioxide emission timeseries over the past few years.*

The version of CEDS emission data we used is the released version for CMIP6. In our data the correction by Feng et al. (2020) is already included.

We have slightly changed the description:

*"In the reference simulation ensemble (AMIP), transient/historical anthropogenic and biomass burning/biofuel emissions as well as other forcings (e.g., concentrations of green-house gases) are prescribed using the CMIP6 emission data (Hoesly et al., 2018) with additional corrections by Feng et al. (2020)."*

*Figure 1: It could be clearer to plot changes in burden in units of mg m−2.*

Done.

*Lines 199-200: The inclusion of biomass burning matches the IPCC definition of "anthropogenic" in a radiative forcing context, so that is more than convenience.*

Thanks for this comment. We have modified the sentence to:

*"Note that the anthropogenic change we define here also includes the contribution from biomass burning emissions, which is consistent with IPCC (Intergovernmental Panel on Climate Change) definition of "anthropogenic" in a radiative forcing context."*

*Line 208: The caption for Figure 1 says that the factor is 5, not 10.*

It should read 5 in the text too. Corrected.

*Line 224: What are those changes due to? Changes in surface winds, or sea ice (or, more directly, open ocean) extent?*

The large interannual variations (from AMIP simulations) are mainly caused by changes in surface winds in combination with sea ice concentration and SST changes. The SST and sea ice concentration are prescribed following the AMIP protocol, but their seasonal and interannual variability still can affect the sea salt/marine organic aerosol production.

We have added the following discussions in section 3.1:

*"Both sea salt and marine organic aerosol emissions are dependent on surface winds, SST, and the ocean fraction. The large inter-annual variations (from AMIP simulations) are mainly caused by changes in surface winds in combination with sea ice concentration and SST changes in*

*different years. For the nudged simulations, since the large-scale winds are constrained and single-year SST and sea ice concentration are used, the changes in sea salt and marine organic aerosol burden are very small."*

*Line 240: "Since carbonaceous aerosols are less hygroscopic compared to sulfate". Add "in the model" because there is a wide spectrum of hygroscopicity for carbonaceous aerosols.*

Thanks for the suggestion. Done.

*Caption of Figure 5: I do not understand the sentence beginning with "Simplified names…". What does it refer to? Perhaps it should be in the caption for Figure 6 instead?*

The figure caption was obsolete and is now corrected.

*Line 321: "(that decreases Nd)". A stabilized sulfate would presumably not decrease Nd, so I suggest rephrasing slightly here.*

The statement (in brackets) is now removed.

*Lines 365-366: Could briefly state that that lack of a change in slope is expected, because the relations shown in Figure 9 characterise clouds, not aerosols.*

Yes, we agree. We have added a sentence:

*"Since the relations shown in Figure 9 characterize cloud properties (instead of the relations between aerosols and clouds), the lack of a change in slope (as in Figure 8) is expected."*

*Figure 12d: I am surprised to see the strong RFari over stratocumulus off the Peruvian coast. I would not have expected a strong aerosol absorption there. Where does it come from?*

This is mainly caused by absorption enhancement caused by the light-absorbing aerosols (including both anthropogenic and biomass burning aerosols) above highly reflective clouds. After a further investigation, we find that the contribution is mainly from the biomass burning aerosols. Figure R2.2 (below) shows the decomposed direct aerosol effect by biomass burning aerosols (E2010BB-E1850), where we clearly see the positive RFari off the Peruvian and Ecuadorian coast. The positive RFari magnitude is slightly weaker but similar to what we see in Figure 12 of the original manuscript.

[Figure]

Figure R2.2: Decomposed annual mean direct aerosol effect due to biomass burning aerosol changes between PI (1850) and PD (2010).

We have added one sentence to explain this:

*"The large positive forcing due to aerosol-radiative interactions off the Peruvian coast is mainly caused by absorption enhancement caused by the light-absorbing aerosols (mainly by biomass burning aerosols) above highly-reflective clouds (Haywood and Shine, 1997)."*

Reference

*Haywood, J.M. and Shine, K.P. (1997), Multi-spectral calculations of the direct radiative forcing of tropospheric sulphate and soot aerosols using a column model. Q.J.R. Meteorol. Soc., 123: 1907-1930. https://doi.org/10.1002/qj.49712354307*

*Line 483: What is meant by "fast processes" here? Is that the same as the "rapid adjustments" of Sherwood et al. (2015 https://doi.org/10.1175/BAMS-D-13-00167.1)?*

Here with "fast processes" we mean processes that don't involve the ocean feedback to the atmosphere that is caused by anthropogenic aerosol forcing. We think our definition of "fast processes" is slightly different from the "rapid adjustment" by Sherwood et al. (2015), which indicates "changes that occur directly due to the forcing, without mediation by the global-mean temperature". As pointed out by that study, the rapid adjustment may include fast changes in SST patterns caused by SST changes in areas with relatively shallow mixed layers. Such rapid adjustments are not considered as fast processes for our case.

The first paragraph in section 7 is revised to:

*"The near surface temperature is strongly affected by the radiative heating/cooling caused directly or indirectly by anthropogenic aerosols through fast processes. The "fast processes" defined here are processes that don't involve the ocean feedback to the atmosphere (in response to the anthropogenic aerosol forcing). Our definition of "fast processes" is slightly different from the "rapid adjustment" by Sherwood et al. (2015), which means "changes that occur directly due to the forcing, without mediation by the global-mean temperature". As pointed out by Sherwood et al. (2015), the rapid adjustment may include fast changes in SST patterns caused by SST changes in areas with relatively shallow mixed layers. In this case, the global mean temperature change is negligible, but these SST changes can significantly affect the cloud and circulation patterns. Such rapid adjustments are not considered as fast processes for our case."*

Reference:

*Sherwood, S. C., Bony, S., Boucher, O., Bretherton, C., Forster, P. M., Gregory, J. M., & Stevens, B. (2015). Adjustments in the Forcing-Feedback Framework for Understanding Climate Change, Bulletin of the American Meteorological Society, 96(2), 217-228.*

*The conclusion does not mention nitrate aerosols. Are there plans to include them in the model at some stage? There are suggestions in the literature they have partly replaced sulfate in some regions, like Europe, and could maintain ERFaer to negative values in the future more globally.*

Yes, the representation of nitrate aerosols has been developed for a newer version of E3SM (Wu et al., submitted). It is similar to the treatment used in CAM5 (Zaveri et al., 2021) and CAM6 (Lu et al., 2021).

The following discussion is added in the conclusion:

*"There are still some missing components or processes in our model. For example, atmospheric chemistry is under-represented in E3SMv1, where we only consider simple sulfur chemistry and the oxidants are prescribed. Related to this simplified treatment, currently nitrate aerosols are not treated in E3SMv1. Previous studies (e.g., Bellouin et al., 2011) have shown that nitrate aerosols have partly replaced sulfate in some regions (e.g., Europe). The overall effective aerosol forcing could maintain negative values in the future, even though some fossil fuel emissions are expected to continuously decrease. The representation of nitrate aerosols has been developed for a newer version of E3SM (Wu et al., submitted), with similar treatment used in CAM5 (Zaveri et al., 2021) and CAM6 (Lu et al., 2021).It would be interesting to investigate the impact of nitrate aerosols on the historical and future ERFaer changes using the new model."*

References

*Zaveri, R. A., Easter, R. C., Singh, B., Wang, H., Lu, Z., Tilmes, S., et al. (2021). Development and evaluation of chemistry-aerosol-climate model CAM5-chem-MAM7-MOSAIC: Global atmospheric distribution and radiative effects of nitrate aerosol. Journal of Advances in Modeling Earth Systems, 13, e2020MS002346. https://doi.org/10.1029/2020MS002346*

*Lu, Z., Liu, X., Zaveri, R. A., Easter, R. C., Tilmes, S., Emmons, L. K., et al. (2021). Radiative forcing of nitrate aerosols from 1975 to 2010 as simulated by MOSAIC module in CESM2-MAM4. Journal of Geophysical Research: Atmospheres, 126, e2021JD034809. https://doi.org/10.1029/2021JD034809*

*Wu, M., Wang, H., Easter, R. C., et al. Development and evaluation of E3SM-MOSAIC: Spatial distributions and radiative effects of nitrate aerosol. Submitted to Journal of Advances in Modeling Earth Systems.*

*Bellouin, N., Rae, J., Jones, A., Johnson, C., Haywood, J., and Boucher, O. (2011), Aerosol forcing in the Climate Model Intercomparison Project (CMIP5) simulations by HadGEM2-ES and the role of ammonium nitrate, J. Geophys. Res., 116, D20206, doi:10.1029/2011JD016074.*